# HINT: Helping Ineffective rollouts Navigate Towards effectiveness

## Abstract

Reinforcement Learning (RL) has become a key driver for enhancing the long chain-of-thought (CoT) reasoning capabilities of Large Language Models (LLMs). However, prevalent methods like GRPO often fail when task difficulty exceeds the model's capacity, leading to reward sparsity and inefficient training. While prior work attempts to mitigate this using off-policy data, such as mixing RL with Supervised Fine-Tuning (SFT) or using hints, they often misguide policy updates. In this work, we identify a core issue underlying these failures, which we term low training affinity. This condition arises from a large distributional mismatch between external guidance and the model's policy. To diagnose this, we introduce *Affinity*, the first quantitative metric for monitoring exploration efficiency and training stability. To improve *Affinity*, we propose HINT: **H**elping **I**neffective rollouts **N**avigate **T**owards effectiveness, an adaptive hinting framework. Instead of providing direct answers, HINT supplies heuristic hints that **guide the model to discover solutions on its own**, preserving its autonomous reasoning capabilities. Extensive experiments on mathematical reasoning tasks show that HINT consistently outperforms existing methods, achieving state-of-the-art results with models of various scales, while also demonstrating significantly more stable learning and greater data efficiency. Code is available on Github[1].

## 1 Introduction

RL methods, particularly GRPO (Shao et al., 2024), play a pivotal role in advancing long CoT reasoning (Wei et al., 2022). By avoiding the instability and overhead of training a separate value model, GRPO leverages group-based reward aggregation to deliver stable and efficient learning signals. Such RL approaches (Ahmadian et al., 2024; Shao et al., 2024; Hu, 2025; Yu et al., 2025) have become a key driver of progress in reasoning ability, enabling models to explore solution paths on verifiable problems. Building on these advances, recent reasoning models such as DeepSeek-R1 (Guo et al., 2025), OpenAI-o1 (Jaech et al., 2024), and Kimi-1.5 (Team et al., 2025) have achieved remarkable performance on complex tasks like mathematical problem solving (Shao et al., 2024)and programming (Jiang et al., 2024).

A critical challenge for GRPO, despite its strong empirical performance, is its tendency to generate sample groups consisting entirely of incorrect answers on tasks whose difficulty exceeds the policy model's evolving capacity (Zhao et al., 2025; Yue et al., 2025). In such cases, the learning process suffers from reward sparsity, where the feedback becomes uniform and uninformative (Yu et al., 2025), ultimately reducing training efficiency and wasting valuable data.

**Leveraging external, off-policy data is a key method for addressing this issue.** This method has been implemented in prior work through two main lines of remedies. (I) **Mixed-policy** (Yan et al., 2025; Zhang et al., 2025a; Fu et al., 2025b): Mixed-policy involves interleaving RL with SFT in a hybrid scheme to stabilize training by leveraging off-policy data. (II) **Using hints** (Li et al., 2025; Liu et al., 2025b; Zhang et al., 2025b): To mitigate reward sparsity and ensure continuous training updates, another common approach is to leverage prompts derived from the ground truth during the rollout phase, guiding the model's exploration along correct trajectories.

---

[1]https://anonymous.4open.science/r/HINT-9DD9/

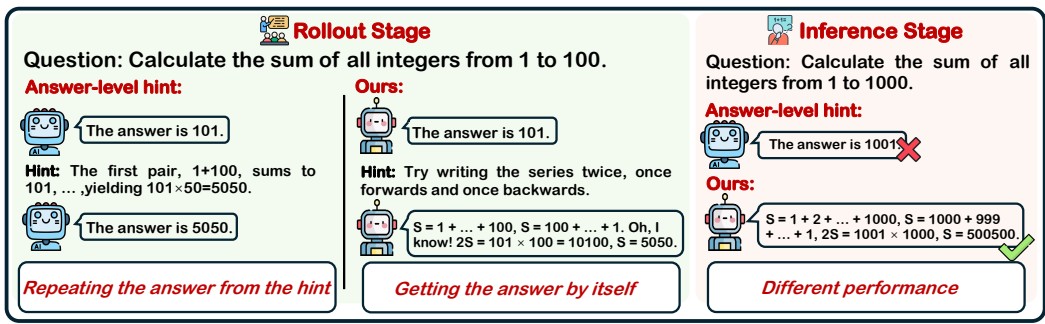

Figure 1: Comparison of Hint Mechanisms and Their Impact on Learning. The answer-level hint provides an explicit partial solution. The model can achieve a reward by simply completing this pre-defined path, which encourages learning a superficial shortcut rather than genuine reasoning. In contrast, our heuristic hint offers a high-level conceptual prompt, **compelling the model to develop its own solution path independently**.

Despite their potential benefits, both of these approaches introduce a significant drawback rooted in a **substantial distributional mismatch**. In mixed-policy training, this mismatch arises between the off-policy SFT data and the on-policy updates, which lead to conflicting gradients and training instability (Yan et al., 2025). Similarly, answer-level hints create a severe mismatch between the distribution of the ground truth and the distribution of the current policy. This results in a deceptive learning signal that, while inflating training rewards, ultimately misguides policy updates toward non-generalizable or spurious solution paths (See Figure 2).

Fundamentally, the aforementioned drawbacks stem from **a lack of what we term training affinity**. This core issue that arises from an **over-reliance** on off-policy sources, such as SFT data or answer-level hints, which inevitably creates a significant distributional mismatch with the model's current policy (Fu et al., 2025a). This mismatch, in turn, leads to excessively high variance in the importance sampling ratios, destabilizing the entire training process. This instability is such a core challenge that prominent algorithms like PPO introduce mechanisms such as clipping to manage it (Schulman et al., 2017), the behavior of which itself provides a signal of training dynamics. To leverage this insight and create a quantitative diagnostic, we define *Affinity* metric in terms of training stability, considering both the frequency of clipping and the variance of the importance sampling ratios.

To leverage off-policy data for enhancing model capability while preserving training affinity, the guiding principle must be to **help the model articulate the solution on its own, rather than being directly told the answer.** To this end, we propose HINT: **H**elping **I**neffective rollouts **N**avigate **T**owards effectiveness, an adaptive hinting framework. As illustrated in Figure 1, HINT implements this principle by providing heuristic hints instead of partial ground-truth answers. These hints serve as high-level guidance, helping the model navigate challenging problems without disclosing solutions. This dynamic is akin to the Socratic method in teaching, where guiding a student with thoughtful prompts, rather than supplying answers, is crucial for developing robust and generalizable reasoning skills.

Our contributions can be summarized as follows:

- We introduce the first formal definition of low training affinity, a key failure mode in RL methods that incorporate off-policy data. Building on this formalization, we propose *Affinity*, a quantitative metric that enables the continuous monitoring of these critical training dynamics.
- To effectively enhance the model's reasoning capabilities while preserving high *Affinity*, we propose HINT, a framework that adaptively providing heuristic hints. HINT guides the model towards successful trajectories without compromising its autonomous exploration and reasoning capabilities.
- Extensive experiments validate our approach. HINT consistently outperforms methods based on mixed-policy and answer-level hints, achieving state-of-the-art results with models of various scales across multiple datasets. Furthermore, our method demonstrates robustness and superior generalization.

## 2 METHODS

### 2.1 PRELIMINARY

Following common practice in recent work, our experiments build on the GRPO algorithm (Guo et al., 2025) while omitting the KL penalty term, as also done in (Yu et al., 2025; Yan et al., 2025). Mathematically, GRPO optimizes the model's behavior through the following objective function:

$$\mathcal{J}_{\mathrm{GRPO}}(\theta) = \mathbb{E}_{(q,a)\sim D,\{o_i\}_{i=1}^G \sim \pi_{\theta_{old}}(\cdot|q)}$$

$$\left[ \frac{1}{G} \sum_{i=1}^G \frac{1}{|o_i|} \sum_{t=1}^{|o_i|} \Big( \min\Big( \frac{\pi_\theta(o_{i,t} \mid o_{i,<t})}{\pi_{\theta_{\mathrm{old}}}(o_{i,t} \mid o_{i,<t})} A_{i,t}, \mathrm{clip}\Big( \frac{\pi_\theta(o_{i,t} \mid o_{i,<t})}{\pi_{\theta_{\mathrm{old}}}(o_{i,t} \mid o_{i,<t})}, 1 \pm \epsilon \Big) A_{i,t} \Big) \Big) \right], \quad (1)$$

for each prompt, GRPO draws a group of $G$ rollouts and computes a group-normalized advantage for every token. Let $\{R_i\}_{i=1}^G$ denote the sequence-level rewards assigned to these rollouts. The token-level advantages $A_{i,t}$ are computed by normalizing each trajectory's reward within the group:

$$A_{i,t} = \frac{R_i - \mathrm{mean}(\{R_j\}_{j=1}^G)}{\mathrm{std}(\{R_j\}_{j=1}^G) + \varepsilon}. \quad (2)$$

When all rollouts in a group are assigned identical rewards, $R_i - \mathrm{mean}(\{R_j\}_{j=1}^G)$ becomes zero for every $i$, causing every advantage $A_{i,t}$ to collapse to zero. Such prompts therefore provide no learning signal during training. Conversely, prompts that produce non-identical rewards across the group yield non-zero advantages and therefore generate meaningful gradients.

### 2.2 THE ILLUSION OF HIGH REWARD

A central challenge in RL is discovering successful trajectories under a limited sampling budget. Although most approaches rely on the reward signal during training to evaluate learning quality, this signal is not always reliable or accurate. To demostrate this, we conduct a simple experiment where we train Qwen2.5-7B (Team, 2024) on the DAPO-Math-170K (Yu et al., 2025), with periodic evaluation on MATH-500 (Hendrycks et al., 2021) test set. During the training phase, if all of its rollouts for a problem are incorrect, we will give an answer-level hint to the model.

Figure 2 shows the outcome of this experiments. Answer-level hint rapidly boosts rewards, creating the illusion of faster convergence. However, the plot on the bottom reveals a different story, as this apparent improvement does not translate into better generalization, with test accuracy stagnating at a low level. Furthermore, providing more detailed hints does not necessarily yield better outcomes, since excessive bias may cause the model's behavior to deviate substantially from its current policy and potentially destabilize training.

The discrepancy between high training rewards and stagnant test accuracy raises a critical question: **why does an apparently strong learning signal fail to produce a generalizable policy?** Our analysis reveals that this problem originates from the severe answer leakage caused by

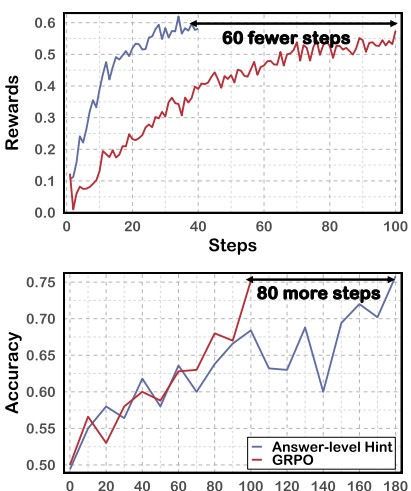

Figure 2: A comparison of training rewards (top) and test accuracy (bottom). High rewards during training do not necessarily lead to high test accuracy, indicating that reward signals may be misleading indicators of model generalization.

answer-level hints. At a mechanistic level, these hints encourage large deviations from the current policy, generating updates with high importance ratios. These updates are then frequently clipped, which nullifies much of the potential learning signal. While this points to the importance of clipping, we find that its frequency alone is an incomplete indicator of training quality. The stability and diversity of the updates that survive clipping are also crucial for effective learning. To properly

diagnose these dynamics, we must quantify both how much of the learning signal survives clipping and the variability of those surviving updates. This motivates our proposal of a new set of metrics to evaluate exploration efficiency and quality.

## 2.3 QUANTIFYING EXPLORATION EFFICIENCY AND QUALITY

The foundation for our new metrics is a direct analysis of the clipping mechanism, which constrains policy updates within a trust region (Schulman et al., 2015). While clipping improves stability, it also suppresses part of the original learning signal, making it difficult to evaluate how effectively the model leverages sampled trajectories. To quantitatively assess this, we focus on two factors that critically influence training quality: (I) the frequency with which policy updates are clipped, and (II) the variability of importance ratios. The first determines how much of the learning signal survives clipping, while the second reflects how stably the surviving updates are distributed. To capture these two aspects in a principled way, we introduce two complementary metrics: Effective Update Ratio (EUR), which measures how much of the learning signal survives clipping, and Update Consistency (UC), which characterizes the stability of the surviving updates. As we will show, these metrics further motivate a unified measure, *Affinity*, which combines both dimensions into a single indicator of exploration efficiency and update quality.

**Effective Update Ratio (EUR).** We use the EUR to quantify **how many token-level updates remain unclipped** under the clipped objective introduced in PPO (Schulman et al., 2017), while staying within the trust-region regime that supports TRPO's monotonic improvement guarantee (Schulman et al., 2015).

Consider a sampled trajectory $(s_1, a_1), \ldots, (s_T, a_T)$. For each token step $i$, let $s_i$ denote the prefix tokens before generation step $i$, and let $a_i$ be the token generated at that step. The policy $\pi_\theta(a_i \mid s_i)$ represents the current model, while $\pi_{\theta_{\text{old}}}(a_i \mid s_i)$ denotes the behavior policy used to collect the trajectory. We define the importance ratio as $r_i = \frac{\pi_\theta(a_i|s_i)}{\pi_{\theta_{\text{old}}}(a_i|s_i)}$ and its log form $\ell_i = \log r_i$, which quantifies the local divergence between the updated policy and the behavior policy. A token is considered to remain within a trust region if its log ratio satisfies $|\ell_i| \leq \delta$, equivalently $e^{-\delta} \leq r_i \leq e^\delta$. The term $A_i$ denotes the token-level advantage, obtained by distributing the trajectory advantage across tokens. Given these definitions, we introduce the EUR:

$$\text{EUR} = \frac{\sum_i w_i \mathbf{1}|\ell_i| \leq \delta}{\sum_i w_i}, \quad w_i = |A_i|, \quad \ell_i = \log \frac{\pi_\theta(a_i \mid s_i)}{\pi_{\theta_{\text{old}}}(a_i \mid s_i)}. \tag{3}$$

EUR measures the advantage-weighted fraction of token-level updates whose probability ratios remain inside the trust region and therefore behave like unclipped PPO updates. This quantity is crucial because unclipped updates preserve the true policy gradient direction, whereas clipped updates either attenuate or nullify it, leading to ineffective learning even when reward appears high.

Importantly, we show that EUR provides (I) a principled estimate of the proportion of gradient contributions that remain unclipped under the PPO surrogate, and (II) a proxy for controlling the upper bound of policy divergence in the sense of the TRPO improvement guarantee. These two facts together imply that a high EUR indicates stable and meaningful policy improvement, while a low EUR signals that most gradient contributions are suppressed and the optimizer is effectively operating with a near-zero learning rate. We present the full derivations and theoretical justification in Appendix A.1.

**Update Consistency (UC).** While EUR measures how many token-level updates remain usable, it does not capture how consistent these effective updates are. In practice, even if a large proportion of updates fall within the trust region, their magnitudes may vary substantially: some updates correspond to very small log-ratios (i.e., conservative steps), while others lie near the trust-region boundary (i.e., aggressive steps). To distinguish stable updates from unstable ones, we introduce the UC metric.

Recall that $\ell_i = \log \frac{\pi_\theta(a_i|s_i)}{\pi_{\theta_{\text{old}}}(a_i|s_i)}$ denotes the log-importance ratio, and $A_i$ the token-level advantage. We focus on the subset of token steps whose updates remain within the trust region, $\mathcal{I} = \{ i : |\ell_i| \leq \delta \}$, which correspond exactly to the unclipped updates in the PPO objective. Within this set, we define the weighted mean log-ratio as $\mu_\ell = \frac{\sum_{i \in \mathcal{I}} |A_i| \ell_i}{\sum_{i \in \mathcal{I}} |A_i|}$. With these quantities in place, we define

the UC as the advantage-weighted standard deviation of the log-importance ratios:

$$\text{UC} = \sqrt{\frac{\sum_{i \in \mathcal{I}} |A_i| \, (\ell_i - \mu_\ell)^2}{\sum_{i \in \mathcal{I}} |A_i|}}. \tag{4}$$

A low UC indicates that the effective updates exhibit small variability in their log-ratios and thus form a stable and coherent update direction. Conversely, a high UC indicates that the supposedly valid updates differ significantly in magnitude, with many lying near the trust-region boundary, which in turn suggests unstable or oscillatory learning dynamics. In other words, while EUR captures the quantity of effective updates, UC captures their quality by measuring whether these updates move the policy in a consistent direction.

We further show in Appendix A.2 that UC is closely related to the variance of the local KL divergence and therefore reflects the stability of the policy update within the trust region. This connection provides the theoretical motivation for using UC alongside EUR to characterize the reliability of gradient-based policy improvement.

**Affinity.** While EUR quantifies how many token-level updates remain effective and UC measures how consistent those effective updates are, neither metric alone is sufficient to characterize the quality of policy improvement. A high EUR may still correspond to unstable learning if the valid updates exhibit large variability (i.e., high UC), indicating that many of them lie near the trust-region boundary and pull the policy in conflicting directions. Conversely, a low UC provides little value when EUR is small, as almost all gradients are clipped and the policy barely changes despite being "consistent".

A desirable training process therefore requires both a substantial number of effective updates (high EUR) and stable, coherent update magnitudes (low UC). To capture this joint requirement in a single measure, we introduce the unified metric *Affinity*. Let $\delta$ denote the log-ratio trust-region threshold used to define the unclipped set in EUR and UC, and let $\tau = \delta/2$ be a temperature parameter controlling the sensitivity of UC. We define:

$$\textit{Affinity} = \text{EUR} \cdot \exp\Big(-\frac{\text{UC}}{\tau}\Big). \tag{5}$$

This multiplicative formulation ensures that *Affinity* is high only when both conditions hold simultaneously: EUR must be large enough to provide meaningful learning signal, and UC must be small enough to guarantee that those updates move the policy in a stable direction. The exponential term modulates the influence of UC, yielding a smooth but decisive penalty on inconsistent updates.

As a result, *Affinity* serves as a holistic indicator of exploration efficiency and training stability in online RL. It summarizes, in a single scalar quantity, both the amount and the quality of effective PPO updates. In Appendix A.3, we further discuss the theoretical motivation for this formulation and its relationship to trust-region optimization principles.

## 2.4 HINT: Helping Ineffective rollouts Navigate Towards effectiveness

The preceding analysis shows that excessively strong or answer-level hints used in prior work tend to degrade training quality by causing frequent clipping (low EUR) or unstable update magnitudes (high UC). **To improve *Affinity*, guidance should therefore avoid providing partial or complete solution steps and instead operate at an abstract, conceptual level that encourages the model to generate the reasoning autonomously.**

We operationalize this principle through the design of HINT. As illustrated in Figure 3, HINT is an adaptive mechanism that **guides the model toward productive reasoning trajectories using hints that are deliberately constrained to the abstract and conceptual level.** These hints avoid revealing answers or intermediate steps and instead provide high-level reasoning cues that activate the model's own problem-solving process, thereby preserving the high-*Affinity* update regime required for stable and effective GRPO training.

Formally, the HINT framework operates as a two-stage process. The first stage mirrors a standard GRPO update cycle. On the rollout stage, for a given problem $q$, the model begins by sampling a set of trajectories $\{o_1, o_2, \ldots, o_G\}$ using its current policy. These trajectories are then evaluated

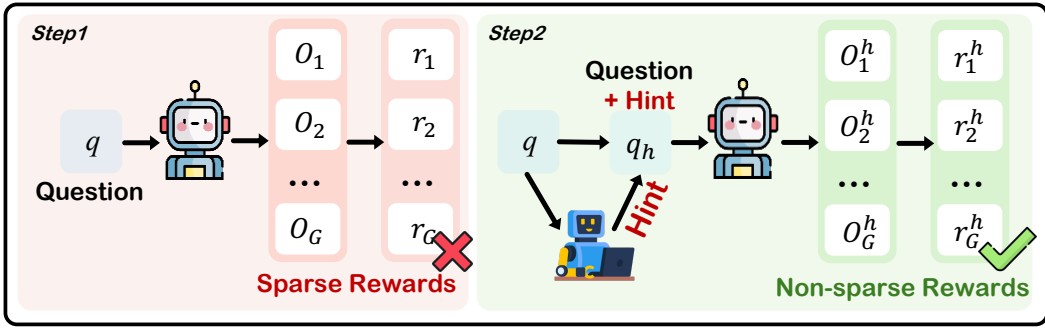

Figure 3: The HINT Framework: An Adaptive Two-Stage Rollout Process. HINT operates in two stages. **(I) Standard Rollout:** The model first samples trajectories from the original problem. If the rewards are non-sparse (at least one is correct), the process follows the standard GRPO update path. **(II) Hint-Augmented Rollout:** If, however, the rewards are sparse (all trajectories are incorrect), the hint mechanism is activated. The model then re-rolls out conditioned on a heuristic hint from a "teacher model". This stage is designed to produce non-sparse rewards, turning a failed sample into a valuable learning opportunity.

by a reward model or predefined rules to obtain a set of rewards $\{r_1, r_2, \ldots, r_G\}$. If these rewards are not sparse (i.e., at least one trajectory is correct), the process proceeds identically to the GRPO algorithm. The non-sparse rewards are used to compute advantages and perform a normal policy update.

The second stage, the hint-augmented rollout, is activated **only if the initial rewards from the first stage are sparse** (i.e., all trajectories are incorrect). In this scenario, where GRPO would stall due to a lack of learning signal, HINT intervenes. A pre-defined hint $h$ is used to construct a hint-augmented query $q_h$. The model is then prompted to resample a new set of trajectories $\{o_1^h, o_2^h, \ldots, o_G^h\}$, this time conditioned on $q_h$. These new, hinted trajectories are re-evaluated to produce a new set of rewards $\{r_1^h, r_2^h, \ldots, r_G^h\}$. This rescue mechanism thus turns a failed rollout into a valuable learning opportunity. By providing a heuristic hint, it is intended to enable a meaningful gradient update, which enhances training efficiency. This is accomplished while the hint itself is carefully constructed to avoid degrading training *Affinity*.

A key design in our method is to decouple the prompts used for trajectory generation from those used for policy optimization. We refer to the input provided to the model during sampling as the rollout prompt, and the input used when updating the policy as the policy prompt.

When HINT is triggered, the rollout prompt may include the hint-augmented version of the problem in order to guide exploration and increase the likelihood of generating successful rollouts. However, the policy prompt is always kept strictly identical to the original problem without any hint. This separation ensures that hints influence only the sampling distribution—not the optimization objective—thereby preventing the model from implicitly learning to rely on hints. In other words, hints are used solely as an exploration aid, while the policy is optimized on the original tasks, preserving the correct training–inference alignment.

## 3 EXPERIMENTS

### 3.1 SETUP

**Experimental Setup.** Our experiments are conducted using Qwen2.5-7B and Qwen2.5-3B (Team, 2024) as backbone models. To ensure a fair and controlled comparison, we constructed a high-quality training set derived from the DAPO-Math-170K dataset (Yu et al., 2025). This process involved using Qwen2.5-72B-Instruct (Team, 2024) to generate four distinct reasoning trajectories for each problem. These outputs were then validated for correctness with Math Verify[2], from which we retained 30k fully correct samples to form our final training data. For baseline methods that

---

[2]https://github.com/huggingface/Math-Verify

require a ground-truth reference solution, we designated the shortest of the four correct trajectories for each problem.

**Benchmarks.** We evaluate the generalization ability of HINT on seven datasets, covering both in-distribution and out-of-distribution scenarios, without using any hint during evaluation. For mathematical reasoning, we adopt AIME24[3], MATH-500 (Hendrycks et al., 2021), OlympiadBench (He et al., 2024), and Minerva (Lewkowycz et al., 2022), which are widely used benchmarks. Since the test sets of AIME24 are relatively small, we report avg@32, while for the other datasets we use pass@1. To assess complex reasoning and out-of-distribution generalization, we further evaluate on ARC-Challenge (Clark et al., 2018), GPQA-Diamond (Rein et al., 2024), and MMLU-Pro (Wang et al., 2024). To demonstrate HINT effectiveness, we conduct systematic experiments across multiple benchmarks.

**Baselines.** We compare HINT against several existing methods designed to improve rollout accuracy rate or rollout efficiency in GRPO. The baselines include: (1)**LUFFY** (Yan et al., 2025): A hybrid approach that combines on-policy and off-policy training, ensuring that each sampled batch contains at least one correct trajectory. (2)**CHORD** (Zhang et al., 2025a): A method dynamically integrating SFT as a weighted objective within on-policy RL. (3)**GHPO** (Liu et al., 2025b): A method that adaptively adjusts the hint length based on the ground-truth solution. If a shorter hint fails to solve the problem, the hint length is progressively increased until the correct answer is obtained. (4)**QuestA** (Li et al., 2025): A method constructs the hint by using the initial 50% of a reasoning trajectory generated by a larger, more capable model. (5)**BREAD** (Zhang et al., 2025b): A binary search–based method that identifies a hint length such that the model's rollouts are neither all correct nor all incorrect, and uses this balanced point as the hint for training.

A comprehensive overview of our experimental configuration, including detailed prompts, hyperparameters, and implementation settings for all methods, can be found in the Appendix B for full reproducibility.

## 3.2 MAIN RESULTS

We benchmarked our proposed method against several mainstream approaches, including both mixed-policy strategies and other hint-based methods. These experiments were conducted on two scales of backbone models: Qwen2.5-7B and Qwen2.5-3B. We report our results in Table 3. Our analysis reveals the following key findings:

**HINT enhances In-Distribution reasoning and teaches problem-solving skills.** HINT significantly enhances the reasoning capabilities of models, achieving state-of-the-art performance on multiple in-distribution benchmarks. Models trained with HINT demonstrate substantial gains, with Qwen2.5-7B and Qwen2.5-3B showing average improvements of 9.0% and 6.8%, respectively, underscoring the effectiveness of our approach. We also observed an interesting emergent behavior during training: when a model encountered two similar, challenging problems, it would often rely on a hint for the first but then solve the second independently by applying the same reasoning pattern. This observation provides strong evidence that our heuristic and minimal hints teach the model how to reason about a class of problems, rather than simply encouraging it to memorize a solution path for a single instance.

**HINT generalizes to Out-of-Distribution problems by optimizing reasoning paths.** HINT also demonstrates strong generalization, enhancing the model's ability to tackle novel problems. Even on out-of-distribution (OOD) test sets, models trained with HINT showed marked improvements. On the OOD test sets, models trained with HINT demonstrated strong generalization, with Qwen2.5-7B and Qwen2.5-3B achieving average performance gains of 7.4% and 1.6%, respectively, highlighting the method's robust ability to generalize. This strong OOD performance is explained by a deeper phenomenon observed in our case studies. We found that the model successfully reapplies high-level reasoning methods from our hints, such as Proof by Contradiction to solve new OOD problems. This demonstrates that our method operates on a conceptual level, effectively teaching the model transferable problem-solving paradigms rather than just answers. It is this acquisition of new, abstract reasoning skills that drives the model's robust generalization.

---

[3]https://huggingface.co/datasets/math-ai/aime24

Table 1: Main Performance Comparison of HINT against Baselines. HINT demonstrates significant performance gains on in-distribution datasets, improving the Qwen2.5-7B and Qwen2.5-3B models by **13.5%** and **6.8%**, respectively. The method also **shows strong generalization capabilities on out-of-distribution data**.

| Methods | In-Distribution | | | | Avg | Out-of-Distribution | | | Avg |
|---|---|---|---|---|---|---|---|---|---|
| | AIME | Math | Olympiad | Minerva | | ARC | GPQA | MMLU | |
| **Qwen2.5-7B** | | | | | | | | | |
| Vanilla | 9.8 | 50.2 | 34.0 | 19.5 | 28.4 | 85.3 | 25.6 | 46.0 | 52.3 |
| GRPO | 12.8 | 75.2 | 40.8 | 31.2 | 40.0 | 87.3 | 30.8 | 56.6 | 58.2 |
| SFT | 13.0 | 77.8 | 42.4 | **32.0** | 41.3 | 77.7 | 25.8 | 44.4 | 49.3 |
| CHORD | 13.2 | 74.4 | 40.0 | 31.2 | 39.7 | 86.6 | 30.1 | 51.2 | 56.0 |
| LUFFY | 12.6 | 70.2 | 38.6 | 30.8 | 38.1 | 87.2 | **32.2** | 46.8 | 55.4 |
| GHPO | 13.1 | 75.6 | 42.2 | 30.0 | 40.2 | 87.0 | 32.0 | 50.0 | 56.3 |
| QuestA | 13.1 | 73.6 | 38.8 | 28.6 | 38.5 | 88.0 | 26.6 | 53.2 | 55.9 |
| BREAD | 11.7 | 72.8 | 41.8 | 29.2 | 38.9 | 85.0 | 29.4 | 48.8 | 54.4 |
| HINT | **13.3** | **79.6** | **43.6** | 31.0 | **41.9** | **88.8** | 31.8 | **58.4** | **59.7** |
| **Qwen2.5-3B** | | | | | | | | | |
| Vanilla | 2.9 | 39.8 | 12.0 | 9.8 | 16.1 | 44.8 | 11.4 | 28.8 | 28.3 |
| GRPO | 4.3 | 44.0 | 18.2 | 12.2 | 19.7 | 45.0 | 11.8 | 28.0 | 28.3 |
| SFT | **5.0** | 48.0 | **20.8** | **14.0** | **22.0** | 20.4 | 7.6 | 20.2 | 16.1 |
| CHORD | 4.5 | 46.6 | 20.2 | 13.0 | 21.1 | 40.0 | 11.0 | 26.4 | 25.8 |
| LUFFY | 3.3 | 40.0 | 18.0 | 13.2 | 18.6 | 40.8 | 11.2 | 24.0 | 25.3 |
| GHPO | 4.0 | 42.2 | 19.6 | 12.8 | 19.7 | 45.5 | **12.0** | 28.2 | 28.6 |
| QuestA | 3.9 | 42.0 | 19.6 | 12.4 | 19.5 | 44.8 | **12.0** | 29.0 | 28.6 |
| BREAD | 4.1 | 44.4 | 20.4 | 13.4 | 20.6 | 45.5 | 11.8 | 29.2 | 28.8 |
| HINT | 4.9 | **48.6** | 20.2 | 13.4 | 21.8 | **48.8** | 11.8 | **30.2** | **29.9** |

**The effectiveness of HINT scales with model size.** Our results show that the benefits of HINT are more pronounced in larger models, with the performance gains for Qwen2.5-7B consistently outpacing those for Qwen2.5-3B across all evaluations. To understand the mechanism behind this trend, we analyzed the training rollouts and found a clear difference in how effectively each model leveraged the provided hints. A quantitative analysis confirmed that out of 100 randomly sampled rollouts where hints were provided to each model, Qwen2.5-7B produced a successful trajectory following the hint 34.0% more often than Qwen2.5-3B did. This superior efficacy in converting hints into successful outcomes directly explains the more pronounced performance gains, indicating that the greater capacity of larger models allows them to better capitalize on the abstract guidance offered by HINT.

## 3.3 Training Dynamics

To investigate the impact of various off-policy strategies, we tracked the EUR, UC, and *Affinity* metrics for our method alongside several key baselines which detailed in Section 3.1, with the full training dynamics plotted in Figure 4. This analysis led to the following key observations.

**In the early stages of training, the model shows strong resistance to off-policy data.** As illustrated in the left plot of Figure 4, all three off-policy methods exhibit a sharp drop in EUR, indicating that clipping occurs very frequently at this stage. We call this initial period the "EUR Collapse Stage", where the model is highly resistant to the off-policy data and the clipping frequency is consequently high. With more training steps, the model gradually adapts, leading to reduced clipping frequency and eventual stabilization. Notably, compared to GHPO and LUFFY, HINT achieves a higher steady-state EUR, demonstrating its superior ability to help the model accommodate and leverage off-policy data.

**Over-reliance on off-policy data often prevents the model from converging.** As shown in the middle plot of Figure 4, both GHPO and LUFFY quickly reach high UC values at the beginning of training and remain at that level. This indicates persistently large variance in importance sampling, which results in unstable model updates and hampers convergence. In contrast, the UC of HINT

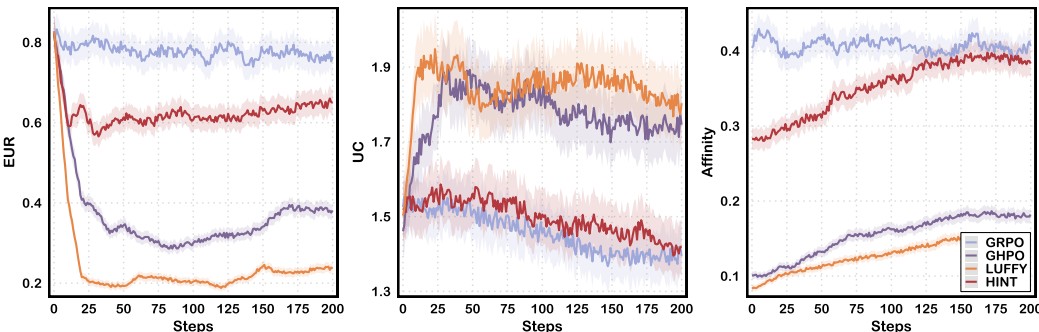

Figure 4: We record the EUR, UC, and *Affinity* metrics across different training processes to investigate the impact of various off-policy strategies on training. **Left:** EUR during training; **Middle:** UC during training; **Right:** *Affinity* during training. Overall, HINT most effectively alleviates the EUR collapse, avoids persistently high UC, and achieves higher *Affinity*, thereby enabling more stable and efficient training.

does not spike early on but instead indicates that our heuristic hints avoid casing large distributional shifts, allowing the policy updates to remain centered around a stable learning direction.

**HINT enables the model to genuinely absorb the knowledge provided by hints.** As presented in the right plot of Figure 4, the *Affinity* of HINT gradually approaches that of GRPO as training progresses. This implies that the model becomes increasingly capable of identifying which hints are truly useful. In other words, HINT enhances training efficiency and sample utilization in the early stages, while maintaining convergence trends consistent with GRPO in the later stages, thereby balancing early gains with eventual stability.

### 3.4 IN-DEPTH ANALYSIS

**Does hinting truly enhance training effectiveness?** We measured the number of valid samples (i.e., rollouts that are not entirely incorrect) generated by GRPO and HINT under an equal computational budget (8 hours of training). As shown in the top of Figure 5, although HINT produced slightly fewer total samples than GRPO, it yielded a greater number of valid samples. This indicates that HINT achieves higher training efficiency under the same time constraints, suggesting that hints guide the model toward more productive exploration trajectories rather than wasting updates on implausible rollouts.

From a broader perspective of the entire training process, the proportion of valid samples with HINT is higher than that of GRPO by 18.9%, further confirming that hinting improves the signal-to-noise ratio of training data. In other words, the gradient updates induced by HINT are more likely to be based on partially correct reasoning chains, thereby amplifying useful supervision signals and mitigating the destabilizing effects of noisy rollouts.

The dominance of valid rollouts under HINT suggests that hints not only improve rollout quality but also reshape the global optimization landscape by steering policy learning toward regions where correct reasoning is more likely to occur. This mechanism explains why HINT can achieve sustained improvements even without relying on answer leakage, ultimately leading to more robust and generalizable training outcomes.

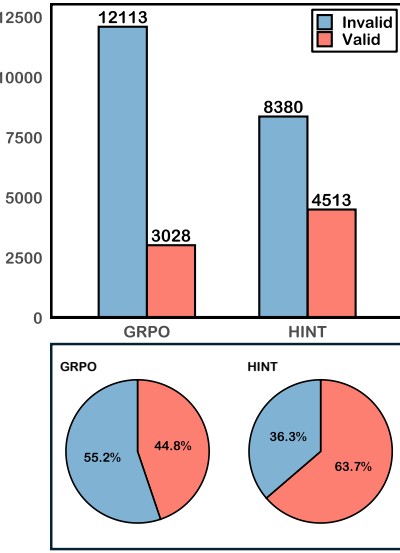

Figure 5: Sampling Efficiency of HINT and GRPO at Different Training Stages. Under an equal budget, HINT yields **1,485 more valid samples** (top) and achieves a **18.9% higher final proportion of valid samples** (bottom).

**Does hinting affect the diversity of model's outputs?**
Entropy serves as a key metric for measuring generation diversity (Cheng et al., 2025; Zheng et al., 2025). Building on the training processes for HINT and the GHPO baseline detailed in Section 3.1, we further compared their dynamics by analyzing the average entropy of reasoning trajectories throughout the training period. For each method, we separately computed the mean entropy on samples with and without hints.

As illustrated in Table 2, on the subset requiring hints, the entropy of HINT is notably higher than GHPO, which is answer-level hints. This is because answer-level hints often provide a "half-completed" reasoning trajectory, forcing the model to follow a predetermined path with limited exploration. In contrast, ours do not disclose specific solution steps, leaving the reasoning process entirely up to the model and thereby encouraging broader exploration across different trajectories.

Table 2: We compare the average entropy for different methods on samples both with and without hints. The results consistently show that **HINT promotes higher entropy than answer-level hints across both scenarios**.

|       | w/ hint | w/o hint | All   |
|-------|---------|----------|-------|
| GRPO  | –       | 0.143    | 0.143 |
| GHPO  | 0.123   | 0.141    | 0.129 |
| HINT  | 0.188   | 0.198    | 0.193 |

Even more surprisingly, we find that on samples where no hints are needed, GHPO still yield the lowest entropy compared to both GRPO and HINT. This suggests that long-term exposure to answer-level hints suppresses diversity at a deeper level: even when no hints are provided, the model's ability to generate diverse reasoning paths is diminished.

## 4    RELATED WORK

**Reinforcement Learning for Large Language Model Reasoning.** Recent advances in RL approaches have significantly enhanced the reasoning capabilities of LLMs. Large reasoning Models (LRMs) such as OpenAI-o1 (Jaech et al., 2024), DeepSeek-R1 (Guo et al., 2025), and Kimi-1.5 (Team et al., 2025) achieve state-of-the-art performance on complex reasoning tasks (e.g., mathematics, coding, scientific problem solving) by leveraging Reinforcement Learning from Verifiable Rewards (RLVR) (Liu et al., 2025a; Hu et al., 2025; Cui et al., 2025), where automatically checkable rules provide supervision signals. Compared to earlier methods like SFT or reinforcement learning from human feedback (RLHF), RLVR has shown superior generalization and robustness (Chu et al., 2025; Snell et al., 2025). Building on this paradigm, subsequent studies have proposed improved optimization strategies and structured prompting techniques that further strengthen reasoning capabilities (Schulman et al., 2017; Wang et al., 2020). Despite this progress, a critical failure mode for existing RL methods is reward sparsity, which occurs when all rollouts in a sample fail. Overcoming this challenge is essential for enhancing the stability and sample efficiency of training large reasoning models.

**Improving Rollout Efficiency in RL for LLMs.** A well-known challenge in methods such as GRPO is the vanishing gradient issue. This problem occurs when all trajectories in a sample group are incorrect, as the group advantage collapses to zero, yielding no gradient for policy updates (Shao et al., 2024; Guo et al., 2025). To mitigate this, some works have focused on injecting external, off-policy data to improve training efficiency and stability. This has been explored through two main strategies. Some methods use mixed-policy, replacing a portion of on-policy rollouts with complete, high-quality trajectories from off-policy datasets (Yan et al., 2025; Lin et al., 2025; Xu et al., 2025; Wang et al., 2025). Others employ partial supervision, providing segments of a ground truth to rescue failed rollouts (Li et al., 2025; Liu et al., 2025b; Zhang et al., 2025b). While these approaches effectively improve rollout efficiency, their over-reliance on off-policy data can misguide policy updates, steering the model toward non-generalizable or spurious solution paths.

## 5    CONCLUSION

In this work, we identify the problem of low training affinity caused by an over-reliance on off-policy data and propose HINT, an adaptive framework to resolve this trade-off. HINT significantly outperforms strong baselines on competitive math benchmarks and demonstrates robust out-of-distribution generalization. Our work showcases a scalable and principled path toward more capable, self-improving reasoning models, with future work pointing towards extending HINT to new domains and modalities.

## 6 ETHICS STATEMENT

This work adheres to the ICLR Code of Ethics. In this study, no human subjects or animal experimentation was involved. All datasets used were sourced in compliance with relevant usage guidelines, ensuring no violation of privacy. We have taken care to avoid any biases or discriminatory outcomes in our research process. No personally identifiable information was used, and no experiments were conducted that could raise privacy or security concerns. We are committed to maintaining transparency and integrity throughout the research process.

## 7 REPRODUCIBILITY STATEMENT

We have made every effort to ensure that the results presented in this paper are reproducible. All code and datasets have been made publicly available in an anonymous repository to facilitate replication and verification. The experimental setup, including training steps, model configurations, and hardware details, is described in detail in the paper.

Additionally, All datasets are publicly available, ensuring consistent and reproducible evaluation results.

We believe these measures will enable other researchers to reproduce our work and further advance the field.

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

APPENDIX

# A  THEORETICAL FOUNDATIONS OF EUR, UC, AND AFFINITY

## A.1  PROOFS FOR EUR

In this section, we provide the theoretical justification for the two main claims made in the main paper regarding the EUR: (I) EUR estimates the fraction of unclipped PPO gradient contributions (Schulman et al., 2017); (II) EUR serves as a proxy for bounding policy divergence in the sense of TRPO's monotonic improvement guarantee (Schulman et al., 2015).

### A.1.1  PRELIMINARIES

For each token step $i$, let

$$r_i = \frac{\pi_\theta(a_i \mid s_i)}{\pi_{\theta_{\mathrm{old}}}(a_i \mid s_i)}, \qquad \ell_i = \log r_i.$$

PPO optimizes a clipped surrogate objective (Schulman et al., 2017), defined as

$$L_{\mathrm{CLIP}}(\theta) = \hat{\mathbb{E}}_i \Big[ \min\big(r_i A_i,\ \mathrm{clip}(r_i, 1-\varepsilon, 1+\varepsilon)\, A_i\big) \Big], \tag{6}$$

and then maximizes $L_{\mathrm{CLIP}}(\theta)$ with respect to $\theta$.

Let $\mathcal{I} = \{i : |r_i - 1| \le \varepsilon\}$ denote the set of unclipped updates and $\mathcal{C}$ the clipped ones. The gradient of equation 6 decomposes as:

$$\nabla_\theta L_{\mathrm{CLIP}} = \mathbb{E}[\nabla_\theta(r_i A_i)\,\mathbf{1}(i \in \mathcal{I})] + \mathbb{E}[\nabla_\theta(r_i^{\mathrm{clip}} A_i)\,\mathbf{1}(i \in \mathcal{C})]. \tag{7}$$

As noted in Schulman et al. (2017), gradients from clipped terms either vanish or are directionally distorted, while terms in $\mathcal{I}$ preserve the correct policy gradient direction.

The Effective Update Ratio is defined in the main paper as:

$$\mathrm{EUR} = \frac{\sum_i |A_i|\, \mathbf{1}(|\ell_i| \le \delta)}{\sum_i |A_i|}. \tag{8}$$

### A.1.2  PROOF OF CLAIM (I): EUR ESTIMATES THE FRACTION OF UNCLIPPED PPO GRADIENT CONTRIBUTIONS

We show that EUR provides a principled empirical estimate of the proportion of gradient contributions arising from unclipped PPO updates. Recall that, for token-level PPO, the unclipped surrogate gradient at position $i$ is

$$g_i = \nabla_\theta(r_i A_i) = A_i\, r_i\, \nabla_\theta \log \pi_\theta(a_i \mid s_i),$$

where $r_i = \frac{\pi_\theta(a_i|s_i)}{\pi_{\theta_{\mathrm{old}}}(a_i|s_i)}$. For updates that fall inside the trust region (i.e., $i \in \mathcal{I}$ with $|\ell_i| \le \delta$), we have $r_i = e^{\ell_i} \approx 1$ because $\ell_i$ is small. Thus the gradient magnitude simplifies to

$$\|g_i\| \approx |A_i|\, \|\nabla_\theta \log \pi_\theta(a_i \mid s_i)\|,$$

and variations in $\|g_i\|$ across token steps are dominated by variations in $|A_i|$. Since $\|\nabla_\theta \log \pi_\theta(a_i \mid s_i)\|$ is locally bounded and does not change substantially across nearby policy iterates, the total contribution of unclipped updates to the overall gradient is proportional to

$$\mathbb{E}[\,|A_i|\, \mathbf{1}(i \in \mathcal{I})\,].$$

Similarly, the total gradient magnitude (including both clipped and unclipped updates) is proportional to $\mathbb{E}[\,|A_i|\,]$. Therefore, the fraction of gradient contributions that originate from unclipped updates is

$$\frac{\mathbb{E}[\,|A_i|\, \mathbf{1}(i \in \mathcal{I})\,]}{\mathbb{E}[\,|A_i|\,]}.$$

By construction, this is exactly the EUR. Consequently, EUR serves as an effective estimator for the fraction of gradient contributions that are not suppressed by clipping.

### A.1.3 Proof of Claim (II): EUR controls policy divergence in the TRPO sense

TRPO (Schulman et al., 2015) establishes a monotonic improvement lower bound dependent on the KL divergence:

$$\eta(\theta) \geq L_{\theta_{\text{old}}}(\theta) - C \cdot D_{\text{KL}}^{\max}(\pi_{\theta_{\text{old}}}, \pi_\theta), \tag{9}$$

where $C$ is a constant depending on $\gamma$ and $\epsilon$. The token-level empirical KL divergence can be approximated by the expectation of log-ratios:

$$D_{\text{KL}}(\pi_{\theta_{\text{old}}} \| \pi_\theta) \approx \mathbb{E}_{s, a \sim \pi_{\text{old}}} [\, |\ell_i| \,].$$

Recall that EUR is the advantage-weighted fraction of updates within the trust region ($|\ell_i| \leq \delta$). Let $\mathcal{C} = \{i : |\ell_i| > \delta\}$ denote the set of clipped updates. The relationship between EUR and the probability mass of $\mathcal{C}$ depends on the distribution of advantages.

**Assumption 1.** *The expected magnitude of advantages for clipped updates is lower bounded by a factor of the global expected magnitude, i.e., $\mathbb{E}[|A_i| \mid i \in \mathcal{C}] \geq \alpha \mathbb{E}[|A_i|]$ for some $\alpha > 0$.*

Under this mild assumption, we can relate EUR to the probability of clipping $P(\mathcal{C})$:

$$1 - \text{EUR} = \frac{\sum_{i \in \mathcal{C}} |A_i|}{\sum_{\text{all}} |A_i|} \approx \frac{P(\mathcal{C}) \cdot \mathbb{E}[|A_i| \mid \mathcal{C}]}{\mathbb{E}[|A_i|]} \geq \alpha P(\mathcal{C}).$$

This implies $P(\mathcal{C}) \leq \frac{1 - \text{EUR}}{\alpha}$. Conversely, the contribution to the KL divergence from clipped samples is lower bounded:

$$D_{\text{KL}} \geq P(\mathcal{C}) \cdot \min_{i \in \mathcal{C}} |\ell_i| > P(\mathcal{C}) \cdot \delta.$$

If EUR is low (close to 0), the advantage mass is concentrated in $\mathcal{C}$. Unless the advantages in $\mathcal{C}$ are negligibly small (contradicting meaningful exploration), a low EUR implies a significant $P(\mathcal{C})$, which forces $D_{\text{KL}}$ to exceed the trust region boundary $\delta$. Therefore, maintaining a high EUR is a necessary proxy for constraining $D_{\text{KL}}$ and preserving the validity of the TRPO bound.

### A.1.4 Summary

Taken together, the results above show that EUR simultaneously quantifies the fraction of gradient mass preserved by the unclipped PPO surrogate and provides a practical handle on the policy divergence term appearing in TRPO's monotonic improvement bound. Consequently, a high EUR indicates that most updates lie within a stable trust-region regime where policy gradients remain informative, whereas a low EUR reveals that clipped updates dominate the optimization process, leading to vanishing effective gradients and ineffective learning.

## A.2 Proofs for UC

In this section, we provide the theoretical justification for the UC metric introduced in the main paper. We show that UC can be interpreted as (I) an advantage-weighted measure of variability in local log-importance ratios among unclipped updates, and (II) a proxy for the variance of the local KL divergence, which is closely tied to the stability of policy updates.

### A.2.1 Preliminaries

Recall that for each token step $i$, we define

$$r_i = \frac{\pi_\theta(a_i \mid s_i)}{\pi_{\theta_{\text{old}}}(a_i \mid s_i)}, \qquad \ell_i = \log r_i,$$

and the trust-region condition $|\ell_i| \leq \delta$ identifies the set of unclipped updates:

$$\mathcal{I} = \{\, i : |\ell_i| \leq \delta \,\}.$$

The token-level advantages are denoted by $A_i$, and we use the absolute values $|A_i|$ as importance weights on the contribution of each token.

Within the set $\mathcal{I}$, we define the advantage-weighted mean log-ratio:

$$\mu_\ell = \frac{\sum_{i \in \mathcal{I}} |A_i| \, \ell_i}{\sum_{i \in \mathcal{I}} |A_i|}, \tag{10}$$

and the UC is given by the advantage-weighted standard deviation:

$$\text{UC} = \sqrt{\frac{\sum_{i \in \mathcal{I}} |A_i|\, (\ell_i - \mu_\ell)^2}{\sum_{i \in \mathcal{I}} |A_i|}}. \tag{11}$$

### A.2.2  UC as a measure of variability among effective updates

As shown in equation 11, UC is precisely the standard deviation of the log-importance ratios $\ell_i$ over the set of effective updates $\mathcal{I}$. A small UC indicates that the $\ell_i$ values within $\mathcal{I}$ are tightly concentrated around their weighted mean $\mu_\ell$, implying that the magnitudes of the effective updates are consistent and that the resulting policy changes are approximately uniform across token positions. In contrast, a large UC reflects substantial variability among the $\ell_i$ values: some effective updates correspond to very small log-ratios (i.e., conservative steps), while others lie close to the trust-region boundary (i.e., aggressive steps). Such heterogeneity results in uneven and potentially unstable policy updates.

Formally, define the normalized weights

$$\tilde{w}_i = \frac{|A_i|}{\sum_{j \in \mathcal{I}} |A_j|}, \quad i \in \mathcal{I}.$$

Then equation 11 can be rewritten as

$$\text{UC}^2 = \sum_{i \in \mathcal{I}} \tilde{w}_i (\ell_i - \mu_\ell)^2,$$

which is the weighted variance of $\ell_i$ under the empirical distribution induced by the advantages $|A_i|$. Thus UC quantifies how "spread out" the log-ratios are among those updates that are not clipped.

### A.2.3  Relation between UC and gradient variance

We now connect UC to the variance of the policy gradient updates. Consider the gradient contribution scale for a single token $i$ within the trust region ($i \in \mathcal{I}$), defined as $X_i = A_i r_i \approx A_i(1 + \ell_i)$. The stability of training depends on the variance of this update scale. Assuming that the advantage $A_i$ and the log-ratio $\ell_i$ are uncorrelated within the local trust region, we can apply the variance decomposition formula $\text{Var}(XY) \approx \mathbb{E}[X]^2\text{Var}(Y) + \mathbb{E}[Y]^2\text{Var}(X) + \text{Var}(X)\text{Var}(Y)$.

Evaluating $\text{Var}(X_i)$ where $X_i \approx A_i + A_i\ell_i$:

$$\text{Var}(g_i) \propto \text{Var}(A_i(1 + \ell_i)) \approx \text{Var}(A_i) + \text{Var}(A_i\ell_i). \tag{12}$$

The first term $\text{Var}(A_i)$ represents the inherent variance of the reward structure (baseline variance), which is irreducible by policy constraint. The second term captures the variance introduced by the policy shift. Applying the decomposition to $A_i\ell_i$:

$$\text{Var}(A_i\ell_i) \approx \mathbb{E}[A_i^2]\text{Var}(\ell_i) + \mathbb{E}[\ell_i]^2\text{Var}(A_i). \tag{13}$$

Inside the trust region, $\ell_i$ is centered near 0, making $\mathbb{E}[\ell_i]^2$ small. Thus, the dominant component of the induced variance is:

$$\text{Var}_{\text{induced}} \approx \mathbb{E}[A_i^2] \cdot \text{Var}(\ell_i).$$

Recall that $\text{UC}^2$ is defined as the advantage-weighted variance of $\ell_i$. Although strictly distinct from the unweighted $\text{Var}(\ell_i)$, they are empirically aligned. As shown in equation 13, UC acts as a multiplicative gain on the gradient variance. A high UC amplifies the gradient noise proportional to the squared advantages $\mathbb{E}[A_i^2]$, destabilizing the update direction. Thus, minimizing UC is theoretically justified to dampen the variance of policy updates specifically arising from diverse importance ratios.

### A.2.4  Relation between UC and local KL variability

We next relate UC to the variability in local KL divergence. The per-state KL divergence between the old and new policy can be expressed as

$$D_{\text{KL}}\big(\pi_{\theta_{\text{old}}}(\cdot \mid s) \,\|\, \pi_\theta(\cdot \mid s)\big) = \mathbb{E}_{a \sim \pi_{\theta_{\text{old}}}(\cdot \mid s)}\big[\log r(a, s)\big].$$

At the token level, the empirical KL is estimated by averaging $\ell_i$ over samples from $\pi_{\theta_{\text{old}}}$. Thus, the variability of $\ell_i$ within $\mathcal{I}$ directly reflects how much the local per-state KL can fluctuate around its mean.

Since TRPO's monotonic improvement bound (Schulman et al., 2015) depends on controlling KL, large fluctuations in $\ell_i$ (i.e., a high UC) indicate that some states experience near-boundary policy changes even when the average KL remains small. This effectively weakens the trust-region assumption and can cause oscillatory learning dynamics. By contrast, a low UC ensures that the per-token KL changes are not only small on average but also uniformly bounded, leading to more reliable surrogate optimization.

### A.2.5 SUMMARY

In summary, UC captures the internal stability of policy updates within the trust region by measuring the advantage-weighted variance of log-importance ratios among unclipped samples. A low UC implies that effective updates move the policy in a coherent and conservative manner, while a high UC reveals that updates, though nominally "valid," are heterogeneous and prone to inducing unstable or oscillatory behavior. Together with EUR, UC thus provides a complementary view of both the quantity and the quality of effective policy updates during training.

### A.3 THEORETICAL DISCUSSION OF AFFINITY

In this section, we provide the theoretical motivation for combining the EUR and UC into the unified *Affinity* metric introduced in the main paper. We show that *Affinity* captures the joint requirements for effective and stable policy updates in PPO-style RL, and we relate its form to principles underlying trust-region optimization.

### A.3.1 PRELIMINARIES

Recall the definitions of EUR and UC from the main paper. Let

$$\ell_i = \log \frac{\pi_\theta(a_i \mid s_i)}{\pi_{\theta_{\text{old}}}(a_i \mid s_i)}$$

denote the log-importance ratio at token step $i$, and let $\mathcal{I} = \{i : |\ell_i| \leq \delta\}$ be the set of unclipped updates under the PPO objective. EUR measures the fraction of effective updates:

$$\text{EUR} = \frac{\sum_i |A_i|\mathbf{1}(i \in \mathcal{I})}{\sum_i |A_i|},$$

while UC quantifies the internal variability of those updates:

$$\text{UC} = \sqrt{\frac{\sum_{i \in \mathcal{I}} |A_i|(\ell_i - \mu_\ell)^2}{\sum_{i \in \mathcal{I}} |A_i|}}, \qquad \mu_\ell = \frac{\sum_{i \in \mathcal{I}} |A_i|\ell_i}{\sum_{i \in \mathcal{I}} |A_i|}.$$

### A.3.2 RATIONALE FOR COMBINING EUR AND UC

As shown in Appendix A.1, EUR provides an unbiased estimate of the proportion of gradient mass preserved by the unclipped PPO surrogate. Hence, a high EUR indicates that most updates meaningfully contribute to the policy gradient. However, EUR alone cannot ensure stability: if the log-ratios within $\mathcal{I}$ vary widely (high UC), many of those "effective" updates may be close to the trust-region boundary and induce oscillatory policy shifts.

Appendix A.2 further shows that UC approximates the variance of token-level policy divergence and characterizes the consistency of unclipped gradients. Yet UC by itself is also insufficient: a perfectly consistent set of updates (low UC) provides little value when EUR is small, since almost all gradients are clipped and the policy barely moves.

Thus, a high-quality update requires satisfying both conditions simultaneously: a sufficiently large proportion of effective updates (high EUR) and low variability among them (low UC).

### A.3.3 AFFINITY AS A JOINT STABILITY-EFFICIENCY INDICATOR

To encode this joint requirement in a single quantity, we define the *Affinity* metric:

$$\text{Affinity} = \text{EUR} \cdot \exp\left(-\frac{\text{UC}}{\tau}\right), \qquad \tau = \delta/2. \tag{14}$$

This multiplicative formulation has two motivations:

**Logical conjunction.** The product structure ensures that a failure in either condition (low EUR or high UC) produces a proportionally low *Affinity*. This structure captures the fact that effective PPO-style updates require both conditions to be satisfied simultaneously, rather than individually.

**Exponential penalty on inconsistency.** Since UC measures weighted variance in log-ratios, an exponential term $\exp(-\text{UC}/\tau)$ acts analogously to an inverse smoothness regularizer, sharply penalizing updates near the trust-region boundary. The temperature term $\tau = \delta/2$ stabilizes the scaling and ensures that the penalty becomes substantial when UC approaches the trust-region limit.

### A.3.4 RELATIONSHIP TO TRUST-REGION OPTIMIZATION

Trust-region methods (including TRPO) rely on bounding the KL divergence to guarantee monotonic policy improvement. While EUR controls the fraction of updates that satisfy the trust-region condition and thus reflects the mean KL contribution, UC characterizes the variability of the local KL divergence within that region by capturing the variance of the log-importance ratios. Consequently, *Affinity* integrates both aspects of policy divergence: high *Affinity* indicates that the empirical KL remains not only small (as ensured by high EUR) but also stable across updates (as ensured by low UC), aligning with the conditions under which trust-region guarantees are most effective.

### A.3.5 SUMMARY

*Affinity* synthesizes two complementary perspectives on PPO update quality: **(I) how many updates remain effective (EUR)**, and **(II) how consistent those updates are (UC)**. The multiplicative formulation in equation 14 captures the synergy required for reliable policy improvement and provides a practical scalar diagnostic for monitoring exploration efficiency and training stability.

## B EXPERIMENTAL DETAILS

### B.1 DETAILED SETUP

**Platform.** All of our experiments are conducted on workstations equipped with 8 NVIDIA A100 PCIe GPUs with 80GB memory.

**Training Data.** The training was performed using a carefully selected subset of the DAPO-Math-170K dataset (Yu et al., 2025). As the original dataset lacks ground-truth solutions, we curated our own by first using Qwen2.5-72B-Instruct to generate four reasoning trajectories for each problem. After validating the final answers with *Math-verify*, we compiled a high-quality training set of 30k problems for which all four generated trajectories were correct. For baselines requiring a ground truth, the most token-efficient of these four correct trajectories was designated as the ground truth. For our methods, we pre-generated the required heuristic hints for the entire 30k-sample training set using Qwen2.5-72B-Instruct. The prompts used in the above process will be detailed in Section B.2.

**Important Parameters of HINT.** HINT is implemented based on the open-source Rl framework lsrl[4]. The RL algorithm employs the GRPO advantage estimator with no KL penalty (kl_coef is set to 0.0). The clipping parameter $\epsilon$ is set to 0.2. For each group, 8 answers are generated, and the training batch size is set to 2. Distributed training utilizes the DeepSpeed library with the *AdamW* optimizer and a learning rate of 1e-6. The *train batch size* is set to 8, *gen batch size* is set to 32, *accum steps* is set to 64, *gen update steps* is set to 128, *temperature* is set to 0.9, *max response* is set to 4096. Mixed-precision training with BF16 is enabled. Memory optimization employs ZeRO Stage 2, with optimizer state offloading to CPU.

---

[4]https://github.com/lsdefine/lsrl

**Important Parameters of Other Baselines.** For baselines with publicly available code repositories, we utilized their official implementations and the parameters specified in their respective publications. For methods without public code, such as BREAD(Zhang et al., 2025b) and QuestA(Li et al., 2025), we reproduced their results using the lsrl framework, strictly adhering to the experimental parameters detailed in their papers.

**Reward Setup.** For our experiments, we employ a sparse, binary reward function. The reward is determined exclusively by the correctness of the final answer in a model's generated trajectory. We use the *Math-Verify* tool for automatic verification, assigning a reward of **+1** for a correct final answer and **0** for an incorrect one.

## B.2 PROMPT LIST

---

**Prompt Template for GRPO**

**System:** You are a helpful AI assistant. A conversation takes place between the User and the Assistant. The User asks a question, and the Assistant solves it. Please help me solve this question. Wrap only the final answer in $\backslash\backslash boxed\{\}$.

**Question:** [Question]

**User:**

---

**Prompt Template for HINT**

**System:** You are a helpful AI assistant. A conversation takes place between the User and the Assistant. The User asks a question, and the Assistant solves it. Please help me solve this question. Wrap only the final answer in $\backslash\backslash boxed\{\}$.

**Hint:** Here are some key information provided to assist you in solving the problem: [Hint]

**Question:** [Question]

**User:**

---

**Prompt Template for Generating hints**

**System:**
* Role and Goal
You are a top-tier problem-solving expert and a master educator. Your goal is not to solve the problem, but to distill the single most critical "Core Insight" or "Aha! Moment" required to find the solution.
* Core Task
You will be given a [Question] and its final [Answer]. Your sole job is to reverse-engineer the most likely solution path and identify the crucial "mental bridge"—the non-obvious insight, change in perspective, or core principle—that unlocks the problem.
* Thinking Framework
Analyze the Gap: First, understand the [Question] and look at the [Answer]. The core difficulty lies in the conceptual space between them. What makes bridging this gap non-

---

trivial? Reconstruct the "Hidden" Step: Mentally construct the most elegant solution path. In that path, what is the single most pivotal, non-obvious leap of logic or application of a principle that a student is most likely to miss? Distill the Insight: Condense this pivotal leap into an extremely short, potent, and core-focused sentence. This sentence is the key that unlocks the door, not the map of the room.

* Constraints

Absolute Brevity: The insight must be a single sentence, ideally under 20 words. No Spoilers: The insight must not reveal any part of the [Answer] or the specific numbers used to calculate it. Inspirational, Not Instructional: It should inspire thought ("heuristic"), not provide a step-by-step recipe ("algorithmic"). Target the Crux: It must address the most critical linchpin that makes the entire solution possible.

* Output Format

Directly output the single, distilled "Core Insight". Do not include any other explanations, headings, or conversational text.

**User:**
### Question:
[Question]
### Answer:
[Answer]

---

Prompt Template for Generating Ground Truth

**System:** You are a helpful AI assistant. A conversation takes place between the User and the Assistant. The User asks a question, and the Assistant solves it. Please help me solve this question. Wrap only the final answer in $\backslash\backslash boxed\{\}$.

**Question:** [Question]

**User:**

---

Prompt Template for Evaluation

**System:** You are a helpful AI assistant. A conversation takes place between the User and the Assistant. The User asks a question, and the Assistant solves it. Please help me solve this question. Wrap only the final answer in $\backslash\backslash boxed\{\}$.

**Question:** [Question]

**User:**

## C    Further Analysis

### C.1    Details of HINT's Entropy

**HINT Encourages Sustained Exploration.** The entropy of the generation distribution serves as a key indicator of exploration diversity. As illustrated in Figure 6, HINT avoids the rapid entropy collapse observed in GRPO during the early stages of training. Instead, HINT maintains a consistently high level of entropy, indicating that the model actively explores when first introduced to the hints. This period of high exploration corresponds directly to the "EUR collapse" phase (discussed in Section 3.3), explaining that while the model initially resists the off-policy guidance, it is nevertheless engaged in a productive and diverse search of the solution space.

During the middle stages of training, HINT's entropy does not decrease monotonically. It exhibits periodic increases. We attribute this to the model encountering novel types of hints and adapting its exploratory behavior to learn how to utilize them. Crucially, even after the policy stabilizes in the later stages, HINT maintains a significantly higher entropy level than GRPO. This provides strong evidence that HINT's heuristic guidance successfully fosters more continuous and diverse exploration, preventing the policy from prematurely converging to a deterministic state.

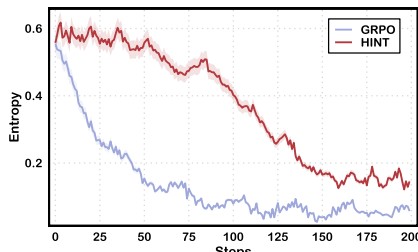

Figure 6: **HINT Prevents Entropy Collapse and Encourages Sustained Exploration.** HINT maintains a high entropy level, especially in the early stages, and stabilizes at a significantly higher value. This demonstrates that HINT's heuristic guidance fosters more continuous and diverse exploration, preventing premature policy convergence.

### C.2    Details of HINT's Accuracy

Our results reveal an interesting trade-off: while the off-policy guidance from HINT may initially slow the rate of convergence, it ultimately enables the model to achieve a higher performance ceiling. As shown in Figure 7, HINT initially exhibits a slower rate of accuracy improvement compared to GRPO. This initial lag is consistent with the early training stages where the model shows resistance to the heuristic hints and has not yet learned to leverage them effectively. However, as training progresses, the model begins to adapt and utilize the guidance. This leads to an accelerated learning rate after approximately 100 steps, with HINT's accuracy eventually surpassing GRPO's and reaching a higher final value. This dynamic suggests that the model requires an adaptation period to master the use of heuristic hints, but once learned, this skill allows it to develop stronger and more robust capabilities.

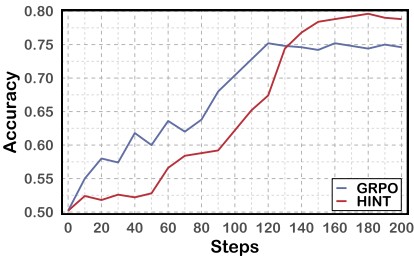

Figure 7: Accuracy of Different Methods. **HINT Achieves Higher Final Accuracy Despite Slower Initial Convergence.**

### C.3    Qwen3-4B Outcome

To verify the effectiveness of HINT on other models, we have supplemented our experiments with results on Qwen3-4B. These results underscore the distinct advantages of HINT in balancing effectiveness and generalization. Compared to the GRPO baseline, HINT delivers comprehensive gains, raising the average scores by 4.1 points on in-domain tasks and 5.6 points on out-of-domain benchmarks, thereby validating the efficacy of our framework. Furthermore, when contrasted with SFT, HINT demonstrates superior robustness; although HINT trails SFT marginally by 0.2 points on in-domain tasks, which is an expected outcome of supervised fitting, it significantly outperforms SFT on out-of-domain benchmarks with a substantial lead of 14.9 points. This stark contrast confirms that while SFT tends to overfit to the domain, HINT cultivates transferable reasoning skills.

Table 3: Main Performance Comparison of HINT against Baselines on Qwen3-4B. HINT consistently outperforms the GRPO baseline, achieving average improvements of **4.1 points** on in-distribution tasks and **5.6 points** on out-of-distribution benchmarks, validating the method's robustness across different model generations.

| Methods | In-Distribution | | | | Avg | Out-of-Distribution | | | Avg |
|---|---|---|---|---|---|---|---|---|---|
| | AIME | Math | Olympiad | Minerva | | ARC | GPQA | MMLU | |
| **Qwen3-4B** | | | | | | | | | |
| Vanilla | 53.3 | 82.0 | 61.8 | 32.2 | 57.3 | 82.5 | 28.0 | 60.4 | 57.0 |
| GRPO | 58.8 | 84.8 | 63.9 | 36.4 | 61.0 | 87.0 | 28.4 | 61.2 | 58.9 |
| SFT | 68.6 | 87.6 | 66.1 | 39.0 | **65.3** | 72.0 | 22.4 | 54.4 | 49.6 |
| LUFFY | 59.6 | 85.2 | 64.4 | 37.1 | 61.6 | 83.0 | 20.1 | 62.2 | 55.1 |
| CHORD | 59.1 | **88.0** | 64.0 | 36.2 | 61.8 | 85.0 | 33.2 | 66.6 | 61.6 |
| GHPO | 62.3 | 87.0 | **66.6** | 34.2 | 62.5 | 87.5 | 29.6 | 68.4 | 61.8 |
| QuestA | 60.3 | 84.4 | 63.0 | 36.6 | 61.1 | **88.4** | 29.6 | 62.3 | 60.1 |
| BREAD | 65.1 | 85.0 | 63.4 | 35.4 | 62.2 | 86.0 | 28.4 | 68.2 | 60.9 |
| HINT | **68.9** | 87.2 | 65.0 | **39.3** | 65.1 | 88.1 | **35.4** | **70.1** | **64.5** |

## C.4 CASE STUDY

---

**Case Study**

**Question** Let $a$ and $b$ be positive integers such that all but 2009 positive integers are expressible in the form $ma + nb$, where $m$ and $n$ are nonnegative integers. If 1776 is one of the numbers that is not expressible, find $a + b$.

**Hint:** How many positive integers cannot be expressed in the form $ma + nb$ with nonnegative integers $m, n$?

**Before the Hint:**

By the two-coin Frobenius result, for co-prime positive integers $a, b$ the number of positive integers not representable as $ma + nb$ equals $\frac{(a-1)(b-1)}{2}$.
So
$$\frac{(a-1)(b-1)}{2} = 2009$$
$$(a-1)(b-1) = 4018$$

Factor $4018 = 41 \cdot 98$. Take $a = 42, b = 99$.
Hence $a + b = 141$.
**Answer:** $\boxed{141}$ ✖

**After the Hint:**

By the two-coin Frobenius result, for co-prime positive integers $a, b$ the number of positive integers not representable as $ma + nb$ equals $\frac{(a-1)(b-1)}{2}$.
So
$$\frac{(a-1)(b-1)}{2} = 2009$$
$$(a-1)(b-1) = 4018$$

Candidates:
$(2, 4019), (3, 2010), (8, 575),$
$(15, 288), (42, 99), (50, 83).$
Require $\gcd(a, b) = 1$,
leaving $(2, 4019), (8, 575), (50, 83)$.
Check 1776: representable for first two, not for $(50, 83)$.
Thus $a + b = 133$.
**Answer:** $\boxed{133}$ ✔

---

**Why the hints works in this question?**

The key to solving this problem is identifying it as an instance of the **Frobenius Coin Problem**. This theorem states that for two coprime positive integers, $a$ and $b$, the number of positive integers not expressible in the form $ma + nb$ is precisely $\frac{(a-1)(b-1)}{2}$. Without guidance, the model correctly sets up the equation $(a-1)(b-1) = 4018$ but may hastily select a factor pair, such as (42, 99), while neglecting the crucial precondition that $a$ and $b$ must be coprime. As shown in the "Before the Hint" example, this oversight leads to an incorrect answer.

The provided hint is designed to steer the model away from this pitfall. By asking about the *number* of non-representable integers, the hint explicitly directs the model's attention toward the Frobenius formula. This encourages a more rigorous, systematic approach: first, finding all possible integer pairs for $(a, b)$; second, filtering these candidates by checking the essential coprimality condition $(\gcd(a, b) = 1)$; and finally, verifying which of the remaining valid pairs satisfies the constraint that 1776 is non-representable. This structured reasoning process, prompted by the hint, is effective because it signals the specific theoretical framework needed to solve the problem, thereby preventing common errors and guiding the model to the correct solution.

C.5 ALGORITHM DETAILS

---

**Algorithm 1** HINT: Helping Ineffective rollouts Navigate Towards effectiveness

---

1: **Input:** initial policy model $\pi_{\theta_{\text{init}}}$; reward models $r_\phi$; task prompts $\mathcal{D}$; hints $\mathcal{H}$; hyperparameters $\epsilon, \beta, \mu$
2: policy model $\pi_\theta \leftarrow \pi_{\theta_{\text{init}}}$
3: **for** iteration = 1, ..., I **do**
4:     reference model $\pi_{\text{ref}} \leftarrow \pi_\theta$
5:     **for** step = 1, ..., M **do**
6:         Sample a batch $\mathcal{D}_b$ from $\mathcal{D}$
7:         Update the old policy model $\pi_{\theta_{\text{old}}} \leftarrow \pi_\theta$
                                        ▷ *Stage 1: Standard Rollout*
8:         Sample $G$ outputs $\{o_i\}_{i=1}^G \sim \pi_{\theta_{\text{old}}}(\cdot \mid q)$ for each $q \in \mathcal{D}_b$
9:         Compute rewards $\{r_{ij}\}_{i=1}^G$ for each $o_i$ by running $r_\phi$
                                        ▷ *Stage 2: Hint-Augmented Rollout (if necessary)*
10:         **if** all rewards $\{r_{ij}\}$ are sparse (e.g., zero) **then**
11:             Get hint $h \in \mathcal{H}$ for problem $q$
12:             Construct hint-augmented query $q_h$
13:             Resample $G$ new outputs $\{o_i^h\}_{i=1}^G \sim \pi_{\theta_{\text{old}}}(\cdot \mid q_h)$
14:             Compute new rewards $\{r_{ij}^h\}_{i=1}^G$
15:             Let $\{o_i\} \leftarrow \{o_i^h\}$, $\{r_{ij}\} \leftarrow \{r_{ij}^h\}$
16:         **end if**
17:         Compute $\hat{A}_{i,t}$ for each token $t$ of $o_i$ using final rewards
18:         **for** HINT iteration = 1, ..., $\mu$ **do**
19:             Update $\pi_\theta$ by maximizing GRPO objective
20:         **end for**
21:         Update $r_\phi$ via replay training
22:     **end for**
23: **end for**
24: **Output:** $\pi_\theta$

---

D  LLM USAGE

Large Language Models (LLMs) were used to aid in the writing and polishing of the manuscript. Specifically, we used an LLM to assist in refining the language, improving readability, and ensuring clarity in various sections of the paper. The model helped with tasks such as sentence rephrasing, grammar checking, and enhancing the overall flow of the text.

It is important to note that the LLM was not involved in the ideation, research methodology, or experimental design. All research concepts, ideas, and analyses were developed and conducted by the authors. The contributions of the LLM were solely focused on improving the linguistic quality of the paper, with no involvement in the scientific content or data analysis.

The authors take full responsibility for the content of the manuscript, including any text generated or polished by the LLM. We have ensured that the LLM-generated text adheres to ethical guidelines and does not contribute to plagiarism or scientific misconduct.

