# OpenReview forum: "HINT: Helping Ineffective Rollouts Navigate Towards Effectiveness"
_ICLR.cc/2026/Conference — ICLR 2026 Conference Withdrawn Submission_

### Official Review · Reviewer_cu2X · 2025-10-17

**Soundness:** 2
**Presentation:** 2
**Contribution:** 2
**Rating:** 2
**Confidence:** 4

**Summary:**

This paper introduced first a new definition of low training affinity, where it is a key failure of RL that incorporate off-policy data. And they introduce a metric that can monitor it during RL training. Furthermore, they introduce a new method that utilizes hints that mimics Socratic method in teaching, where it provides thoughtful prompts rather than partial answers. Results demonstrated its robustness and better generalization.

**Strengths:**

- writing mostly clear, topic is relevant

**Weaknesses:**

- I dont see too much of a difference of this than other methods that also provide hints, such as what if you just give the answer to the model and let it reason, which is what STaR did with rationalization and also [1].
- the method can be summarized to use a teacher model to rewrite the prompt, which the novelty can be an issue overall since many papers did similar things such as STaR with rationalization
- the results seem very weird, for 7b models, for the average scores, almost all methods that leverage off policy data is worse than GRPO, which is not really what the other papers reported in their papers. The 3b model similar trend was found.

[1] GHPO: Adaptive Guidance for Stable and Efficient LLM Reinforcement Learning

**Questions:**

- is the hints/question pair fixed thoughout training?
- which teacher model was used for the experiments?

---

> ### Author Response · Authors · 2025-11-24
> **Response to Reviewer cu2X (1/2)**
>
> We appreciate your critical questions regarding the novelty of our method compared to prior work like STaR, the baseline performance trends, and the experimental setup.
> We have addressed these points in detail below.
>
> # Novelty and Differentiation from STaR and GHPO (Response to Weaknesses 1 & 2)
>
> You suggests that HINT is similar to STaR or simply uses a teacher to rewrite prompts.
> We respectfully argue that there are fundamental structural and functional differences between HINT and methods like STaR or GHPO.
>
> STaR:
> Relies on providing the ground-truth final answer to generate reasoning chains for Supervised Fine-Tuning (SFT).
> It functions as a retrospective correction using the known answer, rather than guiding the model's autonomous exploration process.
>
> GHPO:
> Utilizes answer-level hints (partial steps of the ground truth).
> This creates a "distributional mismatch" (data leakage) that artificially inflates rewards during training without improving actual generalization.
>
> HINT (Ours):
> Employ abstract, conceptual hints (heuristic nudges) within an RL framework, regulated by a novel Affinity metric.
> **This aligns the guidance with the model's policy, effectively steering exploration without leaking the answer or causing distribution shift.**
>
> Beyond the hinting mechanism, a core contribution to our work is the proposal of Affinity (EUR and UC).
> This is the first quantitative metric designed to **diagnose why off-policy data (like that used in mixed-policy methods or using answer-level hints) fails to train base models effectively.**
> It provides a new dimension for monitoring training stability that standard reward curves cannot capture.
>
> # Clarification on Baseline Performance (Response to Weakness 3)
>
> You observed that baselines (methods utilizing off-policy data) performed worse than GRPO, which seemed "weird."
> We clarify that the underperformance of baselines relative to GRPO is attributable to three specific standardizations in our experimental setup that differ from the original papers: (1) Model Choice, (2) Unified Training Dataset, and (3) Unified Training Steps.
> These choices were made to ensure a rigorous and fair comparison, which in turn highlights the robustness of HINT.
>
> - Model Choice (Base vs. Instruct/Math):
> We deliberately utilized Qwen2.5-7B (Base) rather than Instruct or Math-specialized version which is used in baselines' paper to ensure a rigorous evaluation.
> Our rationale is twofold:
> **First, using Base models isolates genuine RL-driven improvements by avoiding the latent bias present in specialized counterparts.**
> As critically noted in Dr.GRPO (Liu et al., 2025) [1], specialized models (e.g., Qwen2.5-Math) often possess latent reasoning patterns or "template-like" behaviors acquired during pre-training.
> These intrinsic capabilities act as confounding factors, making it difficult to distinguish true RL gains from the mere elicitation of pre-existing "SFT-like" skills.
> By using the base model, we eliminate this ambiguity, ensuring that performance gains are strictly attributable to our method.
> Second, base models offer a stricter test of training stability because they are inherently more sensitive to fluctuations in the training process.
> Compared with instruction-tuned models, base models lack the additional alignment and regularization signals that typically improve robustness, making them more prone to instability or collapse during optimization.
> Consequently, demonstrating stable training and strong performance on a base model provides more compelling evidence that HINT effectively alleviates the low-stability issues observed in standard GRPO training.
>
> - Unified Training Dataset:
> To ensure fairness, we did not use the disparate datasets from each baseline paper.
> Instead, we constructed a high-quality, controlled training set of 30k fully correct samples derived from DAPO-Math-170K.
> This ensures that all methods are evaluated on the exact same data distribution.
>
> - Unified Training Steps:
>  We standardized the training budget to 200 steps for all methods to ensure a fair comparison of sample efficiency.
> As seen in Figure 4, methods like GHPO and LUFFY suffer from "EUR collapse" (severe clipping) in the early stages.
> Given a fixed budget of 200 steps, these methods often fail to recover from this initial instability, leading to final results that are inferior to the stable (albeit slower) GRPO.
>
> [1] Liu et al., Understanding R1-Zero-Like Training: A Critical Perspective, COLM 2025

---

> ### Author Response · Authors · 2025-11-24
> **Response to Reviewer cu2X (2/2)**
>
> # Experimental Details (Response to Questions 1 & 2)
> Q1:
> Is the hints/question pair fixed thoughout training?
>
> A1:
> Yes, the hint/question pairs are pre-generated and fixed throughout the training process.
> There is a one-to-one mapping between a problem and its heuristic hint.
> However, our core finding is that the quality and type of the hint matter more than the mere presence of external data.
> As demonstrated in our experiments, fixed heuristic/high-level hints (HINT) significantly outperform fixed answer-level/partial-solution hints (Baselines) because they foster autonomous reasoning rather than pattern matching.
>
> Q2:
> Which teacher model was used for the experiments?
>
> A2:
> As detailed in the Experimental Setup (Section 3.1 and Appendix B.1), we used Qwen2.5-72B-Instruct as the teacher model.
> We selected this model because its strong reasoning capabilities ensure the generation of accurate ground-truth trajectories and high-quality heuristic hints for the training set.

---

### Official Review · Reviewer_n2ue · 2025-10-24

**Soundness:** 3
**Presentation:** 3
**Contribution:** 2
**Rating:** 4
**Confidence:** 3

**Summary:**

This paper proposes a RL framework that leverages hints to handle harder problems where GRPO fails to generate any successful responses, resulting in zero rewards and undefined advantages. The proposed method first generates hints for training problems using a more capable model (e.g. Qwen2.5-72B-Instruct) and fine-tunes a less capable model (e.g. Qwen2.5-7B) with these generate hints as guidance. The proposed method is evaluated on math reasoning benchmarks and outperforms other hint-based RL algorithms.

**Strengths:**

- Extensive experiments with diverse baseline methods substantially strengthens the paper’s claims.
- Overall, the paper is well-written and easy to understand.

**Weaknesses:**

- The reliability of the evaluation for Qwen2.5-7B is questionable, since the majority of hint-based baselines underperform GRPO, despite GRPO not leveraging any hints.
- The idea of using hints is not particularly novel. STaR first introduced answer-level hints [1], and at least two papers on solution-level hints have been accepted to this year’s NeurIPS [2,3]. Actually, most hint-based methods are conceptually similar to jump-start RL [4], in which a guide policy first reduces the search space and then the base policy solves the problem.
- Since the hints are generated by a more capable model, a fair comparison should include a knowledge distillation experiment. However, the paper does not report such results.

[1] Zelikman et al., STaR: Bootstrapping Reasoning With Reasoning, NeurIPS 2022 \
[2] Moon et al., Learning to Better Search with Language Models via Guided Reinforced Self-Training, NeurIPS 2025 \
[3] Zhang et al., BREAD: Branched Rollouts from Expert Anchors Bridge SFT & RL for Reasoning, NeurIPS 2025 \
[4] Uchendu et al., Jump-Start Reinforcement Learning, ICML 2023

**Questions:**

Since most hint-based baselines were released only in mid-2025, I appreciate that the authors have reproduced and reported them. Given that RL performance can vary substantially depending on factors such as the PPO clipping coefficient, the handling of truncated samples, and the specific RL library used, it is challenging to reproduce the baselines exactly. Therefore, I do not place much weight on the precise numerical comparison with these baselines. However, I believe that comparisons with GRPO or knowledge distillation should be conducted more carefully. I will raise the score to 6 or 8 if you answer my questions well.

- The naive knowledge-distillation baseline should (1) generate candidate solutions with Qwen2.5-72B-Instruct, (2) discard unsuccessful responses, and (3) fine-tune Qwen2.5-7B on the remaining successful responses. You may also (4) apply GRPO on top of this baseline. Could you run these experiments?
- You claim that GRPO suffers from entropy collapse. Have you tested recent techniques to address this? For example, a higher clipping ratio [1], Dr-GRPO–style loss aggregation [2], or excluding truncated samples from training [3].
- I believe that the form of the hint plays a critical role. A hint may take various forms, for instance, a partial solution, a concise text extracting essential information from the solution, or a human-written complete solution. It appears that you chose the second form. Could you discuss the advantages and disadvantages of each design choice? You don't have to do additional experiments for this question.

[1] Yu et al., DAPO: An Open-Source LLM Reinforcement Learning System at Scale, arXiv 2025 \
[2] Liu et al., Understanding R1-Zero-Like Training: A Critical Perspective, COLM 2025 \
[3] Luo et al., DeepCoder: A Fully Open-Source 14B Coder at O3-mini Level, 2025

**Details Of Ethics Concerns:**

I have no ethical concerns

---

> ### Author Response · Authors · 2025-11-24
> **Response to Reviewer n2ue (1/4)**
>
> Thank you for your detailed assessment and constructive feedback.
> We particularly appreciate your recognition of our efforts in reproducing baselines, as well as your insightful recommendations regarding the experimental design.
> Following your suggestions, we have conducted additional experiments on knowledge distillation and entropy collapse.
> Our detailed responses are provided below.
>
> # Clarification on Baseline Performance (Response to Weakness 1)
>
> We clarify that the underperformance of baselines relative to GRPO is attributable to three specific standardizations in our experimental setup that differ from the original papers: (1) Model Choice, (2) Unified Training Dataset, and (3) Unified Training Steps.
> These choices were made to ensure a rigorous and fair comparison, which in turn highlights the robustness of HINT.
>
> - Model Choice (Base vs. Instruct/Math):
> We deliberately utilized Qwen2.5-7B (Base) rather than Instruct or Math-specialized version which is used in baselines' paper to ensure a rigorous evaluation.
> Our rationale is twofold:
> **First, using Base models isolates genuine RL-driven improvements by avoiding the latent bias present in specialized counterparts.**
> As critically noted in Dr.GRPO (Liu et al., 2025) [1], specialized models (e.g., Qwen2.5-Math) often possess latent reasoning patterns or "template-like" behaviors acquired during pre-training.
> These intrinsic capabilities act as confounding factors, making it difficult to distinguish true RL gains from the mere elicitation of pre-existing "SFT-like" skills.
> By using the base model, we eliminate this ambiguity, ensuring that performance gains are strictly attributable to our method.
> Second, base models offer a stricter test of training stability because they are inherently more sensitive to fluctuations in the training process.
> Compared with instruction-tuned models, base models lack the additional alignment and regularization signals that typically improve robustness, making them more prone to instability or collapse during optimization.
> Consequently, demonstrating stable training and strong performance on a base model provides more compelling evidence that HINT effectively alleviates the low-stability issues observed in standard GRPO training.
>
> - Unified Training Dataset:
> To ensure fairness, we did not use the disparate datasets from each baseline paper.
> Instead, we constructed a high-quality, controlled training set of 30k fully correct samples derived from DAPO-Math-170K.
> This ensures that all methods are evaluated on the exact same data distribution.
>
> - Unified Training Steps:
>  We standardized the training budget to 200 steps for all methods to ensure a fair comparison of sample efficiency.
> As seen in Figure 4, methods like GHPO and LUFFY suffer from "EUR collapse" (severe clipping) in the early stages.
> Given a fixed budget of 200 steps, these methods often fail to recover from this initial instability, leading to final results that are inferior to the stable (albeit slower) GRPO.
>
> [1] Liu et al., Understanding R1-Zero-Like Training: A Critical Perspective, COLM 2025

---

> ### Author Response · Authors · 2025-11-24
> **Response to Reviewer n2ue (2/4)**
>
> # Novelty and Comparison with Prior Work (Response to Weakness 2)
>
> We appreciate you referencing relevant works like STaR, Jump-Start RL, and BREAD.
> While HINT shares the broad motivation of "guided learning," it differs fundamentally in mechanism and objective.
> We summarize the distinctions as follows:
>
> - STaR [1] and SFT-based methods:
>
>   - These methods rely on "rationalization" or "hindsight," providing the model with the ground-truth answer to retrospectively generate reasoning chains, which are then used for Supervised Fine-Tuning (SFT). **The model tends to memorize specific solution templates ("SFT memorizes") rather than developing robust exploration strategies, often leading to poor out-of-domain generalization.**
>
> - Jump-Start RL [4] and Guided Reinforced Self-Training [2]:
>
>   - These methods typically employ a fixed guide policy to initialize trajectories or narrow the search space, often handing over control to the base policy after a few steps. They treat guidance primarily as an initialization tool rather than a dynamic correction mechanism for distribution alignment. **They do not explicitly address the quality of the gradient update (Affinity), risking instability if the guide and base policies diverge significantly.**
>
> - BREAD [3] and Partial-Solution methods:
>
>   - These approaches use "expert anchors" or partial solution steps to branch rollouts, forcing the model to complete a specific path. As detailed in our "Forms of Hints" analysis, forcing a model to follow an expert's partial trace creates a "broken" reasoning chain (distribution mismatch). **This restricts exploration diversity and can confuse the model due to style differences between the anchor and its internal policy**.
>
> - HINT (Our Framework):
>
>   - HINT strictly withholds answers and partial solutions. It provides prospective, heuristic hints (conceptual scaffolding), guiding the direction of exploration rather than forcing a specific path. **This approach ensures the model constructs the solution autonomously, preserving high Affinity (updates remain within the trust region).** It resolves the trade-off between exploration and stability, fostering generalized reasoning skills that transfer to novel tasks.
>
> [1] Zelikman et al., STaR: Bootstrapping Reasoning With Reasoning, NeurIPS 2022
>
> [2] Moon et al., Learning to Better Search with Language Models via Guided Reinforced Self-Training, NeurIPS 2025
>
> [3] Zhang et al., BREAD: Branched Rollouts from Expert Anchors Bridge SFT & RL for Reasoning, NeurIPS 2025
>
> [4] Uchendu et al., Jump-Start Reinforcement Learning, ICML 2023

---

> ### Author Response · Authors · 2025-11-24
> **Response to Reviewer n2ue (3/4)**
>
> # Knowledge Distillation (SFT) Experiment (Response to Weakness 3 & Question 1)
>
> Following your suggestion, we have added a Knowledge Distillation (SFT) baseline to ensure a fair comparison.
> We generated candidate solutions using Qwen2.5-72B-Instruct, filtered for correctness, and fine-tuned the Qwen2.5-7B and 3B base models on these trajectories, to further demonstrate the robustness of our findings, we extended this comparative experiment to the Qwen3-4B model.
> We compared SFT baseline with GRPO and HINT:
>
> | | AIME | Math | Olympiad | Minerva | Avg | ARC | GPQA | MMLU | Avg |
> | :--- | :--- | :--- | :--- | :--- | :--- | :--- | :--- | :--- | :--- |
> | **Qwen2.5-7B** | | | | | | | | | |
> | Vanilla | 9.8 | 50.2 | 34.0 | 19.5 | 28.4 | 85.3 | 25.6 | 46.0 | 52.3 |
> | GRPO | 12.8 | 75.2 | 40.8 | 31.2 | 40.0 | 87.3 | 30.8 | 56.6 | 58.2 |
> | SFT | 13.0 | 77.8 | 42.4 | **32.0** | 41.3 | 77.7 | 25.8 | 44.4 | 49.3 |
> | HINT | **13.3** | **79.6** | **43.6** | 31.0 | **41.9** | **88.8** | **31.8** | **58.4** | **59.7** |
> | **Qwen2.5-3B** | | | | | | | | | |
> | Vanilla | 2.9 | 39.8 | 12.0 | 9.8 | 16.1 | 44.8 | 11.4 | 28.8 | 28.3 |
> | GRPO | 4.3 | 44.0 | 18.2 | 12.2 | 19.7 | 45.0 | **11.8** | 28.0 | 28.3 |
> | SFT | **5.0** | 48.0 | **20.8** | **14.0** | **22.0** | 20.4 | 7.6 | 20.2 | 16.1 |
> | HINT | 4.9 | **48.6** | 20.2 | 13.4 | 21.8 | **48.8** | **11.8** | **30.2** | **29.9** |
> | **Qwen3-4B** | | | | | | | | | |
> | Vanilla | 53.3 | 82.0 | 61.8 | 32.2 | 57.3 | 82.5 | 28.0 | 60.4 | 57.0 |
> | GRPO | 58.8 | 84.8 | 63.9 | 36.4 | 61.0 | 87.0 | 28.4 | 61.2 | 58.9 |
> | SFT | 68.6 | **87.6** | **66.1** | 39.0 | **65.3** | 72.0 | 22.4 | 54.4 | 49.6 |
> | HINT | **68.9** | 87.2 | 65.0 | **39.3** | 65.1 | **88.1** | **35.4** | **70.1** | **64.5** |
>
> **The results reveal a distinct trade-off inherent to SFT, where it exhibits exceptional efficacy on in-distribution tasks but suffers from severe brittleness in out-of-distribution scenarios, a limitation that HINT successfully overcomes.**
>
> # Analysis of Entropy Stabilization and Performance (Response to Question 2)
> We first clarify that mitigating entropy collapse is a secondary benefit of HINT rather than its primary contribution, which is why our initial analysis was placed in the Appendix.
> However, valuing your insightful suggestion to benchmark against recent techniques, we conducted an additional experiment applying the higher clipping ratio strategy ($\epsilon=0.2 \to 0.28$) adapted from DAPO to the Qwen2.5-7B GRPO baseline.
>
> In terms of training dynamics, this adjustment effectively alleviated the entropy collapse typically observed in the early-to-mid stages of standard GRPO training.
> By the later stages, the entropy stabilized at approximately 0.2, eventually surpassing the level maintained by HINT (~0.17). However, despite this sustained exploration diversity, the performance on test benchmarks did not show a corresponding significant uplift compared to the standard GRPO baseline.
> As shown in the table below, while "GRPO + higher clipping" yields marginal fluctuations, its overall average scores remain comparable to standard GRPO and significantly lag behind HINT.
>
> | | AIME | Math | Olympiad | Minerva | Avg | ARC | GPQA | MMLU | Avg |
> | :--- | :--- | :--- | :--- | :--- | :--- | :--- | :--- | :--- | :--- |
> | Vanilla | 9.8 | 50.2 | 34.0 | 19.5 | 28.4 | 85.3 | 25.6 | 46.0 | 52.3 |
> | GRPO | 12.8 | 75.2 | 40.8 | 31.2 | 40.0 | 87.3 | 30.8 | 56.6 | 58.2 |
> | GRPO + clip higher | 12.0 | 76.4 | **43.4** | **31.4** | 40.8 | 87.0 | 30.4 | 56.2 | 57.9 |
> | HINT | **13.3** | **79.6** | 43.6 | 31.0 | **41.9** | **88.8** | **31.8** | **58.4** | **59.7** |
>
> **We conclude that while a higher clipping ratio successfully mitigates the symptom of entropy collapse, it does not translate into substantial gains in model capability.**
> This finding suggests that maintaining high entropy via looser constraints may simply result in "diverse but low-quality" sampling, where the model explores broadly but ineffectively.
> In contrast, HINT distinguishes itself by actively guiding the exploration toward high-reward regions via Affinity-aware hints, thereby delivering tangible and robust performance improvements.

---

> ### Author Response · Authors · 2025-11-24
> **Response to Reviewer n2ue (4/4)**
>
> # Comparison of Hint Forms (Response to Question 3)
> We fully share your view that the specific form of a hint serves as a critical design variable.
> Drawing from both our theoretical affinity analysis and empirical observations, **we establish a hierarchy of effectiveness where heuristic hints outperform partial solutions, which in turn outperform complete solutions.**
>
> We contend that providing complete solutions, whether generated by humans or AI, induces the most severe distribution mismatch despite offering maximum information.
> Under such guidance, the model tends to bypass the reasoning process in favor of mere copying.
> **This severe distribution mismatch triggers massive token clipping within the optimization objective, thereby preventing the model from obtaining effective gradient updates.**
>
> Similarly, partial solutions such as those used in BREAD or GHPO can effectively narrow the search space, but may also constrain the model to a rigid trajectory that does not align well with its internal representations.
> **This misalignment can lead to a disjointed reasoning chain and severely limits the diversity of exploration strategies.**
>
> In contrast, the heuristic hints implemented in HINT strike the optimal balance.
> By providing high-level strategic guidance in the form of a core insight while strictly withholding execution details, HINT compels the model to bridge the gap autonomously.
> **This approach preserves the agency of the model and ensures that policy updates remain within the trust region to maintain high Affinity. Furthermore, it fosters the development of generalized reasoning skills.**
> We have incorporated this comparative discussion into the revised manuscript to provide a comprehensive evaluation of hint design.

---

> ### Comment · Reviewer_n2ue · 2025-11-24
> **Good rebuttal**
>
> Thank you for the detailed explanation to my questions. I believe that the SFT experiment will make your paper stronger. To better distinguish your paper from BREAD or GHPO, it would be helpful to emphasize more clearly in the main text that your method utilizes abstract, conceptual hints. In the current manuscript, I think this point is still not stated clearly.
>
> One minor concern is that there is no ablation that directly shows abstract hints are better than partial solutions. In particular, it would be very convincing to see an experiment where you keep everything in your current method fixed but replace abstract hints with partial solutions and compare the performance. This will clarify whether the performance gains come from the use of abstract hints, rather than from other implementation details.
>
> Since all of my questions have been addressed, I am updating my overall score from 4 to 6 and contribution score from 2 to 3. If time allows, I hope you can include the ablation study. I also wish you constructive conversations with the other reviewers.

---

> > ### Author Response · Authors · 2025-11-27
> >
> > Thank you very much for your thoughtful follow-up and for updating your scores.
> > We appreciate your recognition of our work and the constructive feedback.
> > To address all of your concerns, we have carried out additional experiments and revised the manuscript to make our methodology and contributions clearer.
> >
> > In particular, **we have updated Section 2.4 to explicitly emphasize that HINT relies on abstract, conceptual hints, rather than answer-level or step-level guidance.**
> > The revised text now clearly highlights that our hints are deliberately designed to avoid leaking solution content and instead provide high-level reasoning cues.
> > This distinction is now made explicit in the main method description to better differentiate our approach from BREAD, GHPO, and other prior methods.
> >
> > Following your suggestion, we also conducted the requested ablation study by keeping the entire HINT pipeline fixed and replacing our abstract hints with partial solutions extracted from the ground-truth reasoning.
> > We evaluated two variants:
> >  (1) using the first 25% of the solution, and
> >  (2) using a partial solution of the same length as the HINT hint.
> >  The results are shown below.
> >
> > | | AIME | Math | Olympiad | Minerva | Avg | ARC | GPQA | MMLU | Avg |
> > | :--- | :--- | :--- | :--- | :--- | :--- | :--- | :--- | :--- | :--- |
> > | **Qwen2.5-7B** | | | | | | | | | |
> > | HINT w/ Partial-Solution Hint (25%) | 13.1 | 73.2 | 36.6 | 28.0 | 37.7 | 85.0 | 28.8 | 54.2 | 56.0 |
> > | HINT w/ Partial-Solution Hint (Equal Length) | 10.6 | 71.0 | 38.0 | 28.4 | 37.0 | 84.6 | 29.0 | 54.6 | 56.1 |
> > | HINT | **13.3** | **79.6** | **43.6** | **31.0** | **41.9** | **88.8** | **31.8** | **58.4** | **59.7** |
> > | **Qwen2.5-3B** | | | | | | | | | |
> > | HINT w/ Partial-Solution Hint (25%) | 4.1 | 44.0 | 20.0 | 12.6 | 20.2 | 44.4 | 11.0 | 28.0 | 27.8 |
> > | HINT w/ Partial-Solution Hint (Equal Length) | 3.9 | 43.2 | 18.2 | 12.6 | 19.5 | 46.2 | 11.2 | 27.6 | 28.3 |
> > | HINT | **4.9** | **48.6** | **20.2** | **13.4** | **21.8** | **48.8** | **11.8** | **30.2** | **29.9** |
> >
> > The results clearly show that abstract, conceptual hints outperform partial-solution hints under identical settings across all model sizes and benchmarks.
> > **This confirms that the performance gains of HINT stem from the abstract hinting mechanism itself, rather than from other implementation details.**
> >
> > Thank you again for the positive feedback and for helping us improve the clarity of the paper.

---

### Official Review · Reviewer_MRbK · 2025-10-29

**Soundness:** 2
**Presentation:** 1
**Contribution:** 2
**Rating:** 2
**Confidence:** 3

**Summary:**

This paper introduces a novel hinting framework called HINT, designed to alleviate the commonly observed issue of reward sparsity in reinforcement learning for large language model reasoning tasks. Instead of providing answer-level hints, HINT utilizes heuristic hints to guide the model towards autonomous reasoning, thereby avoiding policy drift and training instability.

**Strengths:**

The authors introduce the novel concept of Training Affinity, along with quantitative metrics—EUR, UC, and Affinity—to evaluate training stability.
- The proposed heuristic hinting mechanism demonstrates advantages over prior methods such as GRPO in various experimental settings.

**Weaknesses:**

- Although the paper dedicates considerable space to the discussion of Affinity and related metrics, these metrics are only used for monitoring during training and are not integrated into the optimization objective. The core contribution still boils down to providing heuristic hints. While these hints appear to yield better scores under the newly proposed metrics, the practical significance of these metrics remains unclear. For example: Why is staying within the trust region inherently good? Why is moderate variability in updates desirable? These assumptions are not justified either theoretically or empirically.
- The writing quality is suboptimal. Many terms are introduced without explanation—for instance, “rollout prompt” and “policy prompt.” While one might infer that these refer to the input during sampling and policy optimization respectively, such terminology is not standard and should be clearly defined. In addition, basic concepts from reinforcement learning and GRPO could be better introduced **before** the methodology section to improve clarity and logical flow.

**Questions:**

- Since HINT may involve two rounds of sampling in the case of sparse rewards, the method could incur higher training time. As shown in Figure 5, under the same time budget, HINT actually produces fewer rollouts. Were the baselines compared under the same number of training steps or the same wall-clock time? If the goal is to demonstrate the efficiency of HINT, comparisons based on fixed time budgets would be more convincing.

---

> ### Author Response · Authors · 2025-11-24
> **Response to Reviewer MRbK (1/2)**
>
> Thank you for the insightful comments regarding the practical significance of our metrics, writing clarity, and training efficiency.
> We have carefully addressed each concern below.
>
> # Practical Significance and Theoretical Justification of Affinity (Response to Weakness 1)
>
> You raised valid points regarding the core contribution of our work and the justification for our metrics.
> We address these as follows:
> We respectfully disagree with the notion that our contribution boils down solely to providing heuristic hints.
> **HINT is not just about "giving a hint", it is a principled framework designed to resolve the conflict between off-policy exploration and on-policy stability, enabling the model to learn from guidance without suffering from distribution collapse.**
> While the hints are the visible mechanism, our fundamental contribution is the Affinity-aware optimization framework.
> We identified the "low training affinity" failure mode in existing mixed-policy methods, where blind injection of off-policy data leads to training instability.
>
> Affinity is not merely a monitoring metric; it serves two fundamental roles that standard methods lack:
> - Defining and Guiding "Good Hints":
> Acting as a feedback monitor, Affinity provides a quantitative definition of an optimal hint.
> **It establishes that a "good hint" is not merely one that leads to the correct answer, but one that enables the model to autonomously construct the solution without causing severe distributional shifts.**
> By guiding us to optimize for high Affinity, HINT ensures that the model receives effective learning signals while maintaining a stable training trajectory, validating our heuristic approach over direct answer-feeding.
> - Superior Diagnostic Capability:
> Affinity accurately exposes training pathologies that standard reward curves often mask.
> While high rewards can create an "Illusion of High Reward", **Affinity reveals the underlying reality: whether the model is actually learning or simply overfitting to a mismatched distribution.**
> It specifically detects when distribution mismatch renders gradient updates ineffective—either by triggering mass clipping (low EUR) or introducing extreme variance (high UC).
> This granular diagnosis allows us to distinguish meaningful policy improvement from superficial reward inflation.
>
> To provide the theoretical grounding you requested, we have extensively revised the Methods section and added a new Appendix A.
> We go beyond heuristic arguments to theoretically justify our assumptions:
> - Trust Region (High EUR):
> In Appendix A.1, we prove that EUR is a necessary proxy for controlling policy divergence.
> Maintaining high EUR bounds the contribution of clipped updates to the KL divergence, preserving the monotonic improvement guarantee of TRPO [1].
> - Moderate Variability (Low UC):
> In Appendix A.2, we use variance decomposition to demonstrate that UC acts as a multiplicative gain on policy gradient variance.
> High UC theoretically amplifies noise from importance sampling, destabilizing the update direction.
>
> [1] Schulman J, Levine S, Abbeel P, et al. Trust region policy optimization[C]//International conference on machine learning. PMLR, 2015: 1889-1897.

---

> > ### Comment · Reviewer_MRbK · 2025-11-24
> >
> > How does affinity guide "good hints"? This aspect does not appear to be reflected in the training pipeline.

---

> > > ### Author Response · Authors · 2025-11-24
> > > **Response: How Affinity Guides the Design of “Good Hints”**
> > >
> > > Thank you for raising this important question.
> > > We clarify that Affinity is not intended to be an explicit optimization term in the training objective.
> > > Instead, **Affinity serves as a principled guideline for defining what constitutes a “good hint” — that is, a form of guidance that enhances exploration while remaining compatible with on-policy optimization.**
> > >
> > > The key role of Affinity lies in shaping the form and strength of the hints.
> > > Our theoretical and empirical analysis demonstrates that certain types of guidance—particularly answer-level or overly strong hints—cause large deviations in probability ratios or unstable update magnitudes.
> > > These effects manifest as low EUR or high UC, both of which compromise the effectiveness of GRPO updates.
> > > Existing hint methods often fail precisely because their hints introduce incompatible signals that lead to persistent instability.
> > >
> > > Based on these insights, HINT deliberately employs conceptual, heuristic hints that provide high-level reasoning cues rather than explicit solution steps.
> > > Such hints stimulate productive exploration while maintaining high Affinity, ensuring that guided rollouts stay within a regime where PPO-style updates remain valid and gradients remain informative.
> > > In this way, **Affinity defines what constitutes a “good hint”: one that expands exploration without inducing destructive update dynamics.**
> > >
> > > Thus, Affinity plays a role analogous to trust-region constraints in TRPO [1].
> > > Although not always inserted as explicit loss terms, these constraints serve as design principles that inform how the update rule should be constructed.
> > > Similarly, Affinity governs the design space of hints to achieve safe, stable, and policy-aligned guidance.
> > > **This mechanism is fully reflected in the training pipeline through the choice of hint form, even though Affinity is not directly optimized during training.**
> > >
> > > [1] Schulman J, Levine S, Abbeel P, et al. Trust region policy optimization[C]//International conference on machine learning. PMLR, 2015: 1889-1897.

---

> > > > ### Comment · Reviewer_MRbK · 2025-11-24
> > > >
> > > > However, the paper appears to directly provide a prompt template without discussing how affinity is incorporated into the design of the prompt. Did you experiment with different templates to analyze variations in affinity? How can one efficiently design prompt templates that improve affinity? I believe this should be a core aspect of your approach, yet there is no mention of it. Therefore, I argue that your primary contribution lies in designing a template that achieves relatively high affinity. If this is the sole focus, the work may lack completeness as a standalone contribution.

---

> > > > > ### Author Response · Authors · 2025-11-25
> > > > >
> > > > > Thank you for these thoughtful follow-up questions.
> > > > > Here, we would like to clarify the completeness of our contribution.
> > > > > While the paper presents a specific hinting template, this template is not the core of our method.
> > > > > **Our work follows a complete conceptual pipeline that begins with a theoretical diagnosis of instability, proceeds to the formulation of Affinity as a quantitative metric, and culminates in the design of a method grounded in this metric.**
> > > > > The Affinity framework thus serves as the foundation upon which HINT is developed.
> > > > > Below we address your concerns in detail.
> > > > >
> > > > > > **Q1:** the paper appears to directly provide a prompt template without discussing how affinity is incorporated into the design of the prompt.
> > > > > >
> > > > > > **Q3:** How can one efficiently design prompt templates that improve affinity?
> > > > >
> > > > > The design of HINT is explicitly driven by the goal of improving Affinity.
> > > > > Our theoretical and empirical analysis shows that answer-level or step-level hints introduce overly strong guidance signals that sharply increase the likelihood of specific solution-aligned tokens.
> > > > > These shifts create large importance-ratio spikes that push many updates outside the trust region, lowering EUR, and they affect different parts of the trajectory unevenly, increasing the variability of update magnitudes and thus raising UC.
> > > > > In other words, answer-level hints consistently give rise to the defining characteristics of low Affinity, namely frequent clipping and unstable updates, and these effects ultimately destabilize PPO/GRPO optimization.
> > > > >
> > > > >
> > > > > **From this analysis, we obtain a practical and principled guideline for designing prompt templates that improve Affinity: a hint should be structured so that the model performs the reasoning by itself, rather than receiving the answer or intermediate steps directly.**
> > > > > Guided by this guideline, we design HINT to use abstract, heuristic hints that provide high-level reasoning cues without revealing solution content.
> > > > > This form of guidance fundamentally differs from prior approaches and directly targets the preservation of high Affinity during training.
> > > > >
> > > > > > **Q2:** Did you experiment with different templates to analyze variations in affinity?
> > > > >
> > > > > Yes, in section 3, our experiments systematically validate both the effectiveness of HINT and the predictive capability of Affinity.
> > > > > HINT achieves the strongest generalization performance across all evaluation sets, outperforming all baselines.
> > > > > More importantly, as shown in Figure 4, HINT maintains the highest Affinity throughout training.
> > > > > **Each baseline corresponds to a distinct hint template, this result demonstrates that Affinity reliably differentiates stable from unstable forms of guidance, and that the conceptual hints used by HINT fall precisely within the class of high-Affinity designs.**
> > > > >
> > > > > > **Q4:** What's our primary contribution?
> > > > >
> > > > > Our contribution goes far beyond providing a prompt template that happens to yield high Affinity.
> > > > > The central contribution of this work is twofold.
> > > > > **First, we are the first to introduce Affinity as a principled and theoretically grounded metric that quantitatively characterizes the stability and effectiveness of policy updates in LLM reinforcement learning.**
> > > > > Affinity reveals why prior hint-based methods are unstable by exposing the low-EUR and high-UC update patterns induced by answer-level or step-level guidance.
> > > > >
> > > > > **Second, Affinity leads directly to a new design principle for constructing hints: an effective hint must enable the model to produce the reasoning autonomously rather than supplying the answer or intermediate steps.**
> > > > > This principle differentiates our approach fundamentally from all existing methods, which either leak partial solutions or rely on step-level guidance that disrupts on-policy optimization.
> > > > >
> > > > > Building on these insights, we develop HINT as a conceptually distinct hinting framework that applies only abstract, high-level guidance to maintain high Affinity and achieve stable learning from external signals.
> > > > > **Together, the progression from identifying the root cause of instability, to formulating a quantitative stability metric, and finally to designing a method grounded in this metric constitutes a coherent and substantial contribution that extends well beyond the design of a single prompt template.**
> > > > >
> > > > > To prevent potential misunderstandings for future readers, we will revise the organization of the paper in the next version.
> > > > > We will make the theoretical diagnosis, the role of Affinity, and the derivation of the hinting principle more explicit so that the connection from theory to method becomes clearer.

---

> > > > > > ### Comment · Reviewer_MRbK · 2025-11-26
> > > > > >
> > > > > > Thank you for your response. I recognize that the heuristic hints you proposed are valuable. However, establishing a pipeline to assess whether a prompt template effectively enhances affinity could be a significant addition to your work. This aspect, however, does not appear to be adequately addressed in your paper, which leaves the work feeling somewhat incomplete. That said, as you have satisfactorily addressed my other concerns, I will raise the rating to 4.

---

> > > > > > > ### Author Response · Authors · 2025-11-27
> > > > > > >
> > > > > > > Thank you for the further clarification and for raising the rating.
> > > > > > > We appreciate your thoughtful feedback.
> > > > > > > Regarding the suggestion to establish a pipeline that can assess, before training, whether a prompt template will effectively enhance Affinity, we would like to clarify an important conceptual point.
> > > > > > >
> > > > > > > Affinity is fundamentally a training-time behavioral metric, analogous to quantities such as entropy.
> > > > > > > **Its value emerges from the interaction between the model’s current policy, the rollout distribution, and the optimization dynamics.**
> > > > > > > As a result, Affinity cannot be reliably predicted a priori purely from the surface form of a prompt template.
> > > > > > > Rather, it must be measured during training, when the model’s evolving policy interacts with the hint type.
> > > > > > > This is precisely why Affinity was introduced as a monitoring metric rather than a pre-training screening tool.
> > > > > > >
> > > > > > > That said, our work does provide a principled qualitative criterion for template design: **hints must avoid leaking answer-level information and should remain at an abstract, conceptual level.**
> > > > > > > This principle is derived from the behavior of Affinity but does not require pre-training evaluation.
> > > > > > > The experiments in Section 3 further substantiate this criterion, showing that our approach maintains consistently higher Affinity during training and achieves superior performance on all evaluation benchmarks.
> > > > > > >
> > > > > > > Thank you again for the constructive feedback and for raising your rating.
> > > > > > > If you have any further questions or suggestions, please feel free to reach out, we would be happy to clarify anything that remains unclear.

---

> ### Author Response · Authors · 2025-11-24
> **Response to Reviewer MRbK (2/2)**
>
> # Writing Quality and Definitions (Response to Weakness 2)
> We have revised the manuscript to ensure rigorous terminology:
> A. Explicit Definitions & Terms
> We have conducted a thorough audit of the Methods section to ensure every term is clearly explained.
> - **We explicitly defined "Rollout Prompt" (sampling input, potentially with hints) and "Policy Prompt" (optimization input, strictly original question) to clarify the decoupling strategy.**
> - We have also clarified basic notations in Section 2 to eliminate ambiguity. For example, in Section 2.3, we explicitly define $ s_i $ as the state (context) and $ a_i $ as the action (token).
> B. Structure Improvement
> **We have added a new "Section 2.1 Preliminary", formally introducing the GRPO algorithm and its core concepts before presenting our methodology.**
> This improves the logical flow and background introduction.
>
> # Training Efficiency and Time Budget (Response to Question 1)
> You asks a crucial question regarding the comparison basis given the sampling overhead.
> We confirm that our analysis in Figure 5 explicitly addresses both perspectives to ensure a comprehensive evaluation:
> **We fixed the wall-clock time to compare efficiency (Top plot), and we fixed the number of training steps to compare sample effectiveness (Bottom plot).**
> - Efficiency under Fixed Time Budget (Figure 5, Top):
> We compared both methods over an identical 8-hour wall-clock period.
> HINT produces fewer total rollouts (12,893) than GRPO (15,141) due to the two-round sampling overhead.
> However, HINT significantly outperforms GRPO in generating valid rollouts (4,513 vs. 3,028).
> This indicates that HINT effectively offsets its computational cost by rescuing failed trajectories, ultimately utilizing GPU time more efficiently to produce valid learning signals per hour.
> - Effectiveness under Fixed Training Steps (Figure 5, Bottom):
> We compared the methods under the same number of total rollouts (200 steps).
> In this setting, HINT demonstrates superior sample efficiency, achieving a valid rollout rate of 63.7%, substantially higher than the 44.8% observed with GRPO.
> This confirms that our framework successfully converts what would otherwise be ineffective compute into effective supervision.

---

### Official Review · Reviewer_dYuh · 2025-10-30

**Soundness:** 3
**Presentation:** 2
**Contribution:** 2
**Rating:** 6
**Confidence:** 2

**Summary:**

This paper addresses a fundamental bottleneck in reinforcement learning (RL) for large language model (LLM) reasoning: reward sparsity when task difficulty exceeds model capacity. The authors identify a novel failure mode—low training affinity—arising from distributional mismatch between off-policy guidance (e.g., hints or supervised fine-tuning data) and the model’s evolving policy. To quantify and mitigate this, they introduce:

Affinity metric, a unified quantitative measure combining Effective Update Ratio (EUR) and Update Consistency (UC) to track exploration efficiency and training stability.

HINT framework (Helping Ineffective rollouts Navigate Towards effectiveness), an adaptive two-stage hinting scheme where heuristic, non-answer-revealing hints are provided only when all sampled trajectories are incorrect. These hints are generated by a stronger teacher model and serve to “rescue” failed rollouts without leaking answers.

**Strengths:**

1. Methodological Soundness

The two-stage adaptive rollout is clean and well-motivated. By activating hints only under reward sparsity, HINT balances guidance and autonomy—avoiding shortcut learning associated with direct answer hints.

2. Empirical Backup

- Comprehensive benchmarks and strong baselines (GRPO, GHPO, LUFFY, etc.) provide convincing evidence.
- The inclusion of Affinity curves (EUR/UC plots) gives transparency to training dynamics and stability analysis.
- Additional analyses (entropy dynamics, sampling efficiency, case studies) strengthen interpretability.

**Weaknesses:**

1. Evaluation

All experiments rely on Qwen2.5 models; testing on a diverse set of latest base models (e.g., Qwen3, Llama) would better demonstrate robustness.

In Figure 4, it seems that GRPO behaves similar to HINT at the middle-late stage of RL training, but GRPO training shows inferior performance boost than HINT. So how is such metrics relevant to the overall performance after training?

2. Theoretical Grounding

The Affinity formulation (Eq. 3) is intuitive but somewhat heuristic—no theoretical derivation linking it to expected return or policy divergence bounds is provided. A more formal justification or ablation could clarify sensitivity.

**Questions:**

What the symbol s_i and a_i mean in Eq. (1)? Though people can guess, but it's better to avoid such guess.

---

> ### Author Response · Authors · 2025-11-24
> **Response to Reviewer dYuh (1/3)**
>
> We sincerely thank the reviewer for the constructive feedback and the detailed assessment of our work.
> We have addressed your concerns regarding model robustness, the correlation between metrics and performance, theoretical grounding, and notation clarifications as follows:
>
> # Robustness on Other Base Models (Response to Weakness 1.1)
>
> To verify the effectiveness of HINT on other models, we have supplemented our experiments with results on Qwen3-4B.
> Furthermore, incorporating the SFT baseline as suggested by Reviewer n2ue, we compared HINT against these methods (following the same setup in Section 3.1), and the results are presented in the table below:
>
> | | AIME | Math | Olympaid | Minerva | Avg | ARC | GPQA | MMLU | Avg |
> |-------------|-------|------|--------|------|------|------|------|------|------|
> | Vanilla | 53.3 | 82.0 | 61.8 | 32.2 | 57.3 | 82.5 | 28.0 | 60.4 | 57.0 |
> | GRPO | 58.8 | 84.8 | 63.9 | 36.4 |61.0 | 87.0 | 28.4 | 61.2 | 58.9 |
> | SFT | 68.6 | 87.6 | 66.1 | 39.0 | **65.3** | 72.0 | 22.4 | 54.4 | 49.6 |
> | LUFFY | 59.6 | 85.2 | 64.4 | 37.1 | 61.6 | 83.0 | 20.1 | 62.2 | 55.1 |
> | CHORD | 59.1 | **88.0** | 64.0 | 36.2 | 61.8 | 85.0 | 33.2 | 66.6 | 61.6 |
> | GHPO | 62.3 | 87.0 | **66.6** | 34.2 | 62.5 | 87.5 | 29.6 | 68.4 | 61.8 |
> | QuestA | 60.3 | 84.4 | 63.0 | 36.6 | 61.1 | **88.4** | 29.6 | 62.3 | 60.1 |
> | BREAD | 65.1 | 85.0 | 63.4 | 35.4 | 62.2 | 86.0 | 28.4 | 68.2 | 60.9 |
> | HINT | **68.9** | 87.2 | 65.0 | **39.3** | 65.1 | 88.1 | **35.4** | **70.1** | **64.5** |
>
> These results underscore the distinct advantages of HINT in balancing effectiveness and generalization.
> Compared to the GRPO baseline, HINT delivers comprehensive gains, raising the average scores by 4.1 points on in-domain tasks and 5.6 points on out-of-domain benchmarks, thereby validating the efficacy of our framework.
> Furthermore, when contrasted with SFT, HINT demonstrates superior robustness; although HINT trails SFT marginally by 0.2 points on in-domain tasks, which is an expected outcome of supervised fitting, it significantly outperforms SFT on out-of-domain benchmarks with a substantial lead of 14.9 points.
> This stark contrast confirms that while SFT tends to overfit to the domain, HINT cultivates transferable reasoning skills.
> We have added these results to Appendix C.3 of the revised manuscript.

---

> ### Author Response · Authors · 2025-11-24
> **Response to Reviewer dYuh (2/3)**
>
> # Relevance of Metrics to Final Performance (Response to Weakness 1.2)
>
> We clarify that the convergence of metrics in the later stages does not imply equivalent learning outcomes.
> Instead, these metrics serve as universal indicators for monitoring the stability and efficiency of the learning process across different RL methods.
> The observation that HINT achieves superior performance despite this late-stage convergence can be explained by the following three aspects:
>
> A. Metrics Definition: Quantifying Learning Efficiency
>
> First, to clarify the metrics defined in Section 2.3:
> - EUR measures the quantity of effective gradient updates.
> - UC measures the consistency of the update direction.
> - Affinity combines both to serve as a holistic proxy for "learning efficiency" and training stability.
>
> B. Convergence Indicates Successful Knowledge Internalization
>
> GRPO, being a strictly on-policy method, naturally maintains high Affinity because all rollouts are generated by the policy itself, without introducing off-policy data or external distribution.
> In contrast, approaches that introduce off-policy data to increase rollout success rates often create substantial distributional mismatch, which in turn significantly reduces Affinity.
> **Crucially, the convergence of HINT's Affinity towards the GRPO baseline serves as strong evidence that the model has successfully internalized the external hints.**
> Unlike typical off-policy methods (e.g., GHPO, LUFFY) that suffer from persistent instability due to "distributional mismatch" (as seen in Figure 4), HINT effectively bridges this gap.
> This indicates that the model is no longer resisting the distributional shift introduced by off-policy data but is utilizing it as naturally as the fully on-policy rollouts inherent to GRPO.
>
> C. Stability without Sacrificing Knowledge Expansion
>
> **HINT achieves a desirable combination that conventional methods fail to obtain: on-policy-level stability together with off-policy-level knowledge expansion.**
> The convergence of the Affinity metrics demonstrates that the external guidance introduced during sparse-reward scenarios is ultimately internalized by the model, allowing the training dynamics to return to a regime comparable to strictly on-policy GRPO despite the temporary use of off-policy signals.
> At the same time, this carefully controlled guidance enables the model to explore high-reward reasoning trajectories that lie beyond the capacity of GRPO alone, effectively expanding the solution space without inducing persistent distribution mismatch.
> In this sense, HINT leverages the benefits of off-policy knowledge injection while preserving the stability traditionally associated with fully on-policy optimization.

---

> ### Author Response · Authors · 2025-11-24
> **Response to Reviewer dYuh (3/3)**
>
> # Theoretical Grounding of Affinity (Response to Weakness 3)
>
> Thank you for pointing out the need for stronger theoretical justification.
> In response, we have extensively revised the Methods section (Section 2) to provide rigorous definitions and added a new Appendix A to formally derive these metrics.
> Specifically, we have established the following theoretical links:
> - EUR & TRPO Guarantee:
> In Appendix A.1, we demonstrate that EUR serves as a principled proxy for controlling the policy divergence.
> We show that under mild assumptions regarding advantage distribution, maintaining a high EUR is necessary to bound the contribution of clipped updates to the KL divergence, thereby preserving the monotonic improvement guarantee of TRPO [1].
> - UC & Gradient Variance:
> In Appendix A.2, we employ variance decomposition to prove that UC (the advantage-weighted variance of log-ratios) acts as a multiplicative gain on the variance of the policy gradient.
> Theoretical analysis shows that high UC amplifies the noise introduced by importance sampling, directly destabilizing the update direction.
> - Affinity as a Unified Proxy:
> Based on these derivations, Affinity is formalized not just as a heuristic product, but as a joint indicator requiring both the quantity of effective signal (high EUR) and the stability of the gradient estimate (low UC).
> We believe these derivations provide the solid theoretical grounding requested.
>
> [1] Schulman J, Levine S, Abbeel P, et al. Trust region policy optimization[C]//International conference on machine learning. PMLR, 2015: 1889-1897.
>
> # Notation Clarification (Response to Question 1)
>
> In our formulation (Eq. 1), the notation follows standard RL terminology for language models:
>
> - $s_i$ denotes the state, which includes the prompt and the sequence of generated tokens up to step $i$.
> - $a_i$ denotes the action, which corresponds to the token generated at step $i$.
>
> Beyond these specific additions, we have conducted a comprehensive review of Section 2 to ensure that every notation and term is explicitly defined and unambiguous for all readers. Thank you for your careful and thorough review.

---

### Note · Authors · 2026-01-06

I have read and agree with the venue's withdrawal policy on behalf of myself and my co-authors.